# Anthropometric and physical fitness indicators in the combine draft between the finalist and the eliminated player in the national basketball association all-star slam dunk contest

**Tse-hau Tong**[1]*, **Guo-wei Wang**[2,3]

**1** Department of Astronautical Science and Mechanics, Harbin Institute of Technology, Harbin, Heilongjiang, China, **2** Strength and Conditioning and Sport Performance Lab, Institute of Physical Education, Zhengzhou University, Zhengzhou, Henan, China, **3** Institute of Physical Education, Zhengzhou University, Zhengzhou, Henan, China

* 610909173@qq.com

## Abstract

Little is known about the difference of anthropometry and physical fitness between the finalist and eliminated player in the NBA all star slam dunk contest. This study aimed to compare the difference on anthropometric and physical fitness indicator in the combine draft between finalist and eliminated player in the national basketball association all star slam dunk contest. Draft data of 32 basketball players (N = 32, age in draft year: 20.69±2.28 years old, height without shoes: 196.75±8.68 cm, weight: 96.85±10 kg, body fat percentage: 6.07 ±1.23%) participating in the 2000–2015 draft and 2003–2023 slam dunk contest was selected from national basketball association database. It was classified into finals group (FG) (N = 16) and elimination group (EG) (N = 16). Independent sample t-test with cohen's d was adopted for evaluating the statistical significance of intergroup difference and its effect size. The result indicates that Finalist group was significant less than elimination group on height without shoes (FG vs EG: 193.43±9.47 cm vs 200.06±6.52 cm, P<0.05), standing reach (FG vs EG: 257.66±12.32 cm vs 268.29±10.03 cm, P<0.05) and weight (FG vs EG: 93.38±7.37 kg vs 100.33±11.25 kg, P<0.05). Conversely, compared to elimination group, finalist group has significant better performance on three quarter court sprint (FG vs EG: 3.15±0.1 s vs 3.26±0.12 s, P<0.05), standing vertical jump (FG vs EG: 84.88±5.13 cm vs 78.83±4.9 cm, P<0.05) and max vertical jump (FG vs EG: 102.39±6.47 cm vs 94.79±8.34 cm, P<0.05). However, effect size analysis indicated that height without shoes, standing reach, weight (cohen's d = 0.73–0.959, 0.7≤cohen's d<1.3, moderate) from the anthropometric indicator and three quarter court sprint, standing vertical jump, and max vertical jump (cohen's d = 0.97–1.21, 0.7≤cohen's d<1.3, moderate) from physical fitness indicator has moderate effect size, whereas effect size of body fat percentage, wingspan and lane agility time (cohen's d = 0.31–0.67, 0.3≤cohen's d<0.7, small) was small. To conclude, specific anthropometric and physical fitness indicator shows clear difference between finals group and elimination group. Height without shoes, standing reach, weight in anthropometry and

**Data Availability Statement:** All data in our study are available from the NBA database, URL:www.nba.com/stats/draft/combine.

**Funding:** The authors received no specific funding for this wor.

**Competing interests:** The authors have declared that no competing interests exist.

three quarter court sprint, standing vertical jump, and max vertical jump in physical fitness are key indicator to slam dunk performance. In line with the result in the study, NBA staff can select suitable rookies for slam dunk contest. Similiarly, coach from NBA or other basketball league, who want to improve the player's slam dunk performance, should use specific training programs to develop the slam dunk-related indicator.

# 1 Introduction

Anthropometric measurement and physical fitness test was used in the National Basketball Association (NBA) combine draft for selecting the potential NBA players [1]. NBA combine draft includes anthropometric measurement, physical fitness test, spot-up shooting test and non-stationary shooting test and scrimmages. All rookies, who participated in the combine draft, must be at least 19 years old, with experience in other basketball league [1, 2]. Draft sequence indicates the rank of rookies, which was selected early or lately by NBA team. Therefore, rookies, who rank in the top sequence, always attract more attention in the future session [3]. In order to diagnose current physical condition as well as to predict the future achievement for athletes, anthropometry and physical fitness test in the combine draft has been extensively applied in the competitive sports [1, 2, 4–6]. Likewise, sports performance in the slam dunk contest may also be explained by anthropometry and physical fitness test in the combine draft.

NBA all star game has great attraction to basketball audience, during all star weekend from 2003 to 2018, with up to totally 659.7 television audiences in 1000 family in Los Angeles, USA [7]. Slam dunk contest was hold on all star weekend, from February to March every year, which contained preliminary and finals. Though all the rookies with less than three year of NBA experience were allowed to apply for the contest, the number of competitors was kept between three and six [8]. All the competitors drew lots to decide their order of play before the preliminary. After one player completed a slam dunk in limit time, each referee should give one individual score from six to ten, while the sum of all referee's score is the total score of one slam dunk. Each referee can give six as individual score, if one competitor failed to make one slam dunk in required time. Competitors, who ranked in top two or top three in preliminary, can reach the finals. When both of two players won the top score (50 points) in the finals, each player needs to perform an extra slam dunk until champion is occurs. If one competitor wants to win the top score, he needs to show aesthetic in the take-off phase and flight phase of one slam dunk, because aesthetic was considered by referees as well as the difficulty in the assessment. As a result, the aesthetic in slam dunk plays a vital role in winning the championship during NBA slam dunk contest, which not exists in the NBA regular season [9].

Therefore, competitors with different sports performance in the NBA slam dunk contest, were likely to own special characteristic on anthropometry and physical fitness. Finalist in the contest probably possesses optimal body fat percentage [1, 2], height and standing reach [1, 2, 10–12], weight [1, 2, 10–12], wingspan [1, 2] in the anthropometry, and extremely outstanding performance on linear sprint [1, 2, 10, 13], standing vertical jump and countermovement jump [11, 14, 15], max vertical jump [1, 2, 11, 14, 15], and reactive strength index (RSI) [16] in the physical fitness test, supporting their slam dunk-specific performance. By contrast, some players were eliminated in the preliminary, because of poor slam dunk performance, which may be resulted from unsuitable anthropometric characteristic and unsatisfactory physical fitness.

Thus, research assessing thebasketball players, needs to propose the novel insights about players in the slam dunk contest to define their own characteristic in anthropometry and physical fitness. These insights allow the basketball coaches from NBA or other basketball league to develop the slam dunk performance of players, based on the slam dunk-related physical fitness, contributing to player's sports confidence [17] and basketball league's audience rating. Identification of potential differences on anthropometric and physical fitness indicator between different performances in the slam dunk contest (finalist vs eliminated player) can inform the slam dunk-related indicator for selecting the suitable participants into the contest, which can maximize its aesthetic [18] and audience rating. Hence, this study aims to compare the anthropometry and physical fitness indicator in the combine draft between eliminated player and finalist. The hypothesis of the present study refers to that finalist was possessed taller height without shoes and standing reach, longer wingspan, and lower body fat percentage with better performance in the physical fitness test, in relative to eliminated player.

## 2 Materials and methods

### 2.1 Study design

The descriptive design was used [12]. Available data from NBA official combine draft database from 2000 to 2015 was analyzed in this study (available at www.nba.com/stats/draft/combine) [19–21]. The raw data that this study used is available in S1 Data.

### 2.2 Participants

The combine draft data of 32 NBA players from NBA official draft database was sampled as participants in the present study(N = 32, age in draft year: 20.69±2.28 years old, height without shoes:196.75±8.68 cm, weight:96.85±10 kg, body fat percentage: 6.07±1.23%)[19–21]. Table 1 displays participants' characteristics.

Inclusion criterion for subjects in this study: The subject has participated in at least once NBA all star slam dunk contest during 2003–2023 season and combine draft during 2000–2015 season. The subject has complete draft data. Exclusion criterion was undrafted player during 2000–2015 season, non-participant of NBA slam dunk contest during 2003–2023 season, and player with incomplete draft data.

### 2.3 Ethical consideration

This study was non-human subject research, suggesting that all the data of this study were available data from the NBA official draft database [19–21]. All subjects in the study participated in the NBA combine draft on the basis of the informed consent.

**Table 1. Participants' characteristics (N = 32).**

| Indicator | Mean | SD |
|---|---|---|
| Age in draft year | 20.69 | 2.28 |
| Height W/O shoes (cm) | 196.75 | 8.68 |
| Weight (kg) | 96.85 | 10 |
| BFP (%) | 6.07 | 1.23 |

SD–standard deviation; Height W/O shoes–Height without shoes; BFP–body fat percentage.

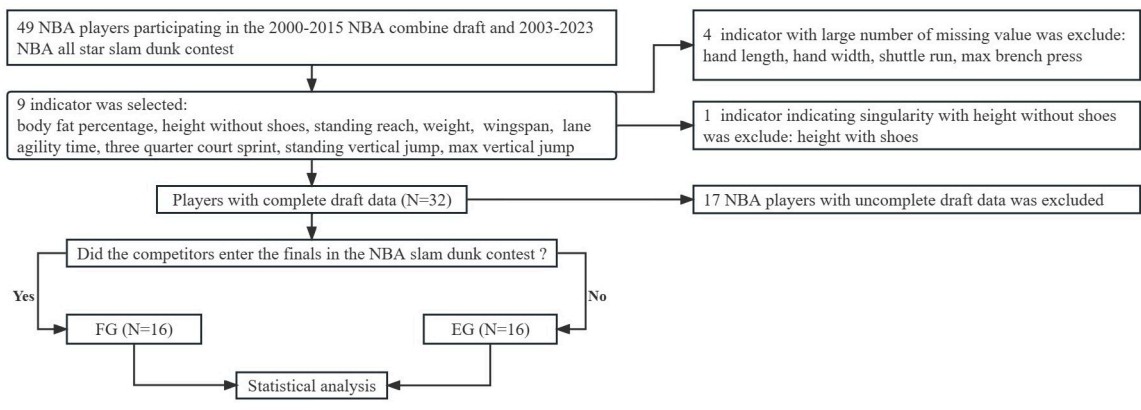

**Fig 1. Experimental procedure of this study.**

## 2.4 Experimental procedure

The experimental procedure in this study was presented in the Fig 1. At first, the invalid indicator was excluded. Criterion for invalid indicator was or extremely correlated with other indicators (r>0.9) and excessive missing rate (missing rate: >50%), indicating the ratio of how many subjects missing the data of this indicator [1]. According to the criterion, height with shoes (r>0.9), which extremely interrelate with height without shoes in previous study [1], hand length (missing rate: >53.06%), hand width (missing rate: >53.06%), in the anthropometry as well as shuttle run (missing rate: >61.22%), max bench press (missing rate: >90%) in the physical fitness test was excluded [19–21]. The reason of excluding the height with shoes rather than height without shoes was that each NBA players owned their unique trainers and shoe pods with various thickness. This may result in the deviation of measurement. Then, the valid indicator was included. Criterion for valid indicator was that indicator has low missing rate (missing rate: <50%) and not extremely interrelated with other indicators (r <0.9). Based on this criterion, body fat percentage (BFP), height without shoes (height W/O shoes), standing reach (SR), weight, wingspan, lane agility time (LAT), three quarter court sprint (TQCS), standing vertical jump (SVJ), and max vertical jump (MVJ) were included as valid indicator in the study (missing rate: 0%, r <0.9).

Subjects with complete data in valid indicator were reserved (N = 32), while those with incomplete data were excluded (N = 17). Based on the replay of 2003–2023 NBA all star dunk contest, all the subject was divided into Finalist Group (FG) (N = 16, BFP: 5.88±0.98%, height:193.43±9.47cm, weight: 93.38±7.37kg), and Elimination Group (EG) (N = 16, BFP: 6.27 ±1.45%, height: 200.06±6.52cm, weight: 100.33±11.25 kg). Inclusion criterion for FG is the competitor, which entered the finals in the slam dunk contest. Inclusion criterion for EG is the competitor, which failed to enter the finals of slam dunk contest. Finally, the statistical analysis was performed, after FG and EG was confirmed.

## 2.5 Measurement of anthropometric and physical fitness indicators

BFP, height W/O shoes, SR, weight, and wingspan were contained in the anthropometric measurement of NBA combine draft. In physical fitness assessment, LAT, TQCS, SVJ, and MVJ were key parameters for linear speed, agility and vertical jump performance in basketball [1]. All the anthropometric measurement and physical fitness test, which exhibits good robustness in the assessment of basketball specific performance, was performed by the NBA conditioning

**Table 2. Definition and test protocol of anthropometric indicators in the NBA combine draft.**

| Indicator | Definition and Test Protocol |
|---|---|
| BFP (%) | BFP is assessed by measuring the skinfold thickness of pectoral, abdomen, and quadriceps using a skinfold caliper. Usually, the right side is usually only measured (for consistency). The tester pinches the skin at the appropriate site to raise a double layer of skin and the underlying adipose tissue, but not the muscle. Then, thecalipers are applied 1 cm below and at right angles to the pinch, and a reading in millimeters (mm) taken two seconds later. In addition, the mean of two measurements should be taken. If the two measurements differ greatly, a third should then be done. Later, the median value can be taken. |
| Height W/O Shoes (cm) | Height W/O Shoes is the measurement the maximum distance from the floor to the highest point of the head, when the subject is facing directly ahead. Shoes should be off, feet together, and arms by the sides. Besides, heels, buttocks and upper back should also be in contact with the wall when the measurement is made. Height W/O Shoes is measured in feet and inches using a physician scale, whereas the player is not wearing shoes. |
| SR (cm) | The player is standing straightly and reaching both arms up, making them vertical to the floor, while their longest tip of their hands to the floor is measured using a measuring tape. |
| Weight (kg) | The player stands with minimal movement with hands by their side. Shoes and excess clothing should be removed. Using a physician scale, body weight is measured. |
| Wingspan (cm) | The tip of the left hand to the tip of the right hand is measured in feet and inches using a measuring tape, while the player is stretching the arms horizontally. |

BFP–body fat percentage; Height W/O Shoes–height without shoes; SR–standing reach.

**Table 3. Definition and test protocol of physical fitness indicators in the NBA combine draft.**

| Indicator | Definition and Test Protocol |
|---|---|
| LAT (s) | A cone is placed at each of four corners of the lane. Starting from the left corner of the free-throw line, the player runs forward to the baseline, side-shuffle to the right corner of the lane, backpedal to the right corner of the free-throw line, and side- shuffle to the left to go back to the starting point. Subsequently, the player changes the direction, side-shuffle to the right corner of the free-throw line, runs forward to the baseline, side-shuffle to the left corner of the lane, and backpedal to go back to the starting point. The score is the time to cover the distance measured in seconds. |
| TQCS (s) | Two cones are placed at corners of the lane along the baseline. Other two cones are placed at the corners of the opposite free-throw line. The player sprints from the baseline to the three-quarter length of the court (22.86 meter) as fast as possible, and their time measured. |
| SVJ (cm) | The player jumps vertically as high as possible and taps the Vertec device without a running start (both feet flat on the floor). The score refers to the difference between the standing reach and jump reach. |
| MVJ (cm) | With running, the player jumps vertically as high as possible and taps the Vertec device. The player can take any number of steps as long as the approach distance is between the free-throw line and a 15-foot (4.6 m) arc, and can also choose either a one-foot or two-foot takeoff. The difference between the standing reach and the maximum jump reach is measured |

LAT–lane agility time; TQCS–three quarter court sprint; SVJ–standing vertical jump; MVJ–max vertical jump.

coach by valid and reliable protocol [1, 22, 23]. The definition and test protocol of anthropometric and physical indicator in the combine draft was demonstrated in Tables 2 and 3.

## 2.6 Statistical analysis

At first, descriptive statistics was adopted for all the data in the study. It was presented as mean and standard deviation (Mean ± SD). Next, Shapiro–Wilks analysis was adopted for testing the normal distribution of data. Levene test was used to examine the homogeneity of variables. The independent sample t-test was employed to test thestatistical significance of intergroup difference, while the data met the normal distribution and the homogeneity of variables.

**Table 4. Comparison of anthropometric and physical indicator in the combine draft between finalist group (N = 16) and elimination group (N = 16) (95%CI).**

| Indicator | FG | | EG | | P |
|---|---|---|---|---|---|
| | Mean | SD | Mean | SD | |
| Anthropometry | | | | | |
| BFP (%) | 5.88 | 0.98 | 6.27 | 1.45 | 0.387 |
| Height W/O Shoes (cm) | 193.43 | 9.47 | 200.06 | 6.52 | 0.028* |
| SR (cm) | 257.66 | 12.32 | 268.29 | 10.03 | 0.012* |
| Weight (kg) | 93.38 | 7.37 | 100.33 | 11.25 | 0.048* |
| Wingspan (cm) | 207.33 | 9.29 | 213.24 | 8.42 | 0.069 |
| Physical fitness test | | | | | |
| LAT (s) | 11.06 | 0.45 | 11.39 | 0.55 | 0.078 |
| TQCS (s) | 3.15 | 0.1 | 3.26 | 0.12 | 0.010* |
| SVJ (cm) | 84.88 | 5.13 | 78.83 | 4.9 | 0.002* |
| MVJ (cm) | 102.39 | 6.47 | 94.79 | 8.34 | 0.007* |

*P<0.05; SD–standard deviation; FG–finalist group; EG–elimination group; BFP–body fat percentage; Height W/O Shoes–height without shoes; SR–standing reach; LAT–lane agility time; TQCS–three quarter court sprint; SVJ–standing vertical jump; MVJ–max vertical jump.

Cohen's d level was employed as effect size to illustrate the extent of intergroup difference and interpreted as: cohen's d<0.3, trivial; 0.3≤cohen's d<0.7, small; 0.7≤cohen's d<1.3, moderate;1.3≤cohen's d<2.1, large; 2.1≤Cohen's d<4.0, very large; cohen's d≥4.0, extreme large [24, 25]. The significance level of normal distribution and homogeneity of variables was set at P>0.05. In addition, the statistical significance of independent sample t-test was set at P<0.05. The statistical analysis was performed using SPSS Statistics 27.0 (IBM Corp, Armonk, New York, USA).

## 3 Results

### 3.1 Anthropometry

The descriptive statistics for anthropometric indicators was displayed in Table 4. Independent sample t-test suggested that FG had significantly lower values than EG on height W/O shoes, SR, and weight (P<0.05), while the intergroup difference on BFP and wingspan were not of statistical significance (P>0.05) (see Table 4). ES test about cohen's d demonstrated that extent of intergroup difference was moderate on height W/O shoes, SR, and weight (0.7≤cohen's d<1.3), while it was small on BFP and wingspan (0.3≤cohen's d<0.7) (see Table 5).

### 3.2 Physical fitness

The descriptive statistics for physical fitness indicators was shown in Table 4. The result of independent sample t-test showed that FG has significant better performance than EG on TQCS, SVJ, and MVJ (P<0.05), while there was no significant intergroup difference on LAT (P>0.05) (see Table 4). ES test about cohen's d suggested that extent of intergroup difference was moderate on TQCS, SVJ and MVJ (0.7≤cohen's d<1.3), whereas it was small on LAT (0.3≤cohen's d<0.7) (see Table 5).

## 4 Discussion

This study found the significant difference on the anthropometry and physical fitness test in the NBA combine draft between FG and EG. It hypothesized that finalist was possessed taller

**Table 5. Effect size level of independent sample t-testbetween finalist group (N = 16) and elimination group (N = 16).**

| Indicator | FG vs EG (95%CI) |
|---|---|
| Anthropometry | |
| BFP (%) | 0.31(-0.93 to 0.46), small |
| Height W/O Shoes (cm) | 0.81(-1.55 to -0.10), moderate |
| SR (cm) | 0.959(-1.66 to -0.200), moderate |
| Weight (kg) | 0.73(-1.52 to -0.07), moderate |
| Wingspan (cm) | 0.67(-1.17 to 0.24),small |
| Physical fitness test | |
| LAT (s) | 0.64(-1.42 to 0.02), small |
| TQCS (s) | 0.97(-1.74 to -0.27), moderate |
| SVJ (cm) | 1.21(0.42 to 1.93), moderate |
| MVJ (cm) | 1.02(0.27 to 1.75), moderate |

FG–finalist group; EG–elimination group; BFP–body fat percentage; Height W/O Shoes–height without shoes; SR–standing reach; LAT–lane agility time; TQCS–three quarter court sprint; SVJ–standing vertical jump; MVJ–max vertical jump.

height without shoes and standing reach, longer wingspan, and lower body fat percentage with better performance in the physical fitness test, in relative to eliminated player. Differently, the result of this study only partly supported the hypothesis, indicating that in TQCS, SVJ, and MVJ performance, FG was significantly superior to EG, while their height W/O shoes, SR, and weight in the anthropometry was significantly less than others.

In the study, anthropometric difference in combine draft was discovered between groups. At first, height and SR was differentiated between FG and EG in anthropometry, suggesting that FG had shorter height and SR than EG in present study. Contrary to the result in current study, Williams MNC et al. reported that national-level players were taller and has taller SR than state-level players, indicating that taller height and SR contribute to on-court performance in basketball players [10]. One possible reason refers to the difference of on-court performance between slam dunk contest and NBA regular season. In slam dunk contest, aesthetic and difficulty of air-technique was key on-court performance, while was not vital to regular season [9]. Especially the aesthetic, it is a determinant of top score in the contest, but is unimportant to score in regular season. In this way, it was likely that FG has better performance than EG on aesthetic and difficulty of air-technique,due to the discrepancies on shorter height and SR between groups. It can be explained by the greater angular velocity. As shown in the formula ($\omega = \frac{v}{r}$), when linear speed (v) is similar between groups, FG will generate greater angular velocity (ω) than EG, because of their shorter arm of force (r), which positively correlated with height and SR (P<0.05) [26]. As a result,greater angular velocity may contribute to the number and velocity of twist, which likely to influence the difficulty and aesthetic of air-technique. For example, short height was extensively found in elite athlete from rhythmic gymnastics. It was highly correlated performance score (P<0.001, r = 0.8), which also scored based on the difficulty and aesthetic of air-technique [18, 27]. In addition,greaterjoint angular velocity, which was resulted from shorter height and SR, may also be benifical for slam dunk-related MVJ. It has been reported that maximal angular velocity of dominant knee extension in take-off phase was positively correlated with spike jump in volleyball (P<0.001, r = 0.85) [28], which has similar jump characteristics (slow stretch-shorten cycle, double legs include lead leg and trail leg, orientation step) with one type of slam dunk-specific MVJ (double-legs

take-off without dribbling) [14, 28, 29]. As a result, if FG finished one slam dunk by this type of MVJ, their flight height will be higher than EG that also used it, which may also lead to the intergroup difference on difficulty and aesthetic of air-technique. However, except height and SR, weight also distinguished between groups in anthropometry, demonstrating that FG possessed less weight than EG in the study. It suggested that FG, who owned the superior slam dunk performance, may has greater lean body mass (LBM) and less fat mass, because its BFP is also less than EG in current study, aside from weight. Similar to the current study, Cui Y et al. Reported that fat mass and LBM had influence on SVJ and MVJ in NBA players ($P<0.01$), which was likely to determine the jump performance on slam dunk [2]. One possible reason was that LBM was positively associated with peak vertical ground reaction force (peak vGRF), peak power, and flight height in countermovement jump (CMJ) ($P<0.001$), while fat mass may have negative association with them [30]. In this way, FG was likely to generate superior peak vGRF and peak power in the take-off phase, which may underpin the flight height of slam dunk-related MVJ (double-legs take-off without dribbling, single-leg with or without dribbling) [11, 29]. Thus, shorter height, shorter SR, and less weight appear to be good anthropometric indicators of sports performance in the slam dunk contest.

Not only anthropometric indicators, but also physical fitness indicators were distinguished between groups. First, in present study, sprint performance in combine draft was differentiated between groups, suggesting that FG has better performance than EG in TQCS, a 22.86-meter linear sprint. Similar to the result in this study, previous studies in first division male basketball players found that, 30 meter sprint (closing to the distance of TQCS) was strongly interrelated with CMJ ($P<0.05$, $r = -0.619$) [13], which might determine the performance on slam dunk-related MVJ in nation-level basketball players ($P<0.05$, $r = 0.51–0.91$) [11, 15]. However, highly correlation between 20–30meter sprint and CMJ was also discovered in track and field athlete ($P<0.05$, $r = -0.73–0.83$) [31, 32]. Therefore, TQCS was likely to support the flight height of slam dunk-related MVJ. The possible reason refers to that TQCS was likely to reflect the ground contact time (GCT) in take-off phase of slam dunk, because both sprinting time (0–30 meter) [33] and the GCT of vertical jump [34, 35] were controlled by reactive strength, also representing the transitional time between eccentric contraction and concentric contraction during SSC [29], and the stiffness of ankle during take-off phase [36]. As the result of different level on reactive strength, long GCT was likely to dissipate more stored elastic energy in eccentric contraction into heat, while short GCT may minimal the energy loss [37]. As a result, FG, who possessed better TQCS than EG, may minmize the GCT in the take-off phase to maximize the flight height of slam dunk, by shortening the transition time and optimising the stiffness of ankle joint during SSC. In this way, it may cause the discrepancy on difficulty and aesthetic of air-technique between FG and EG. In addition, except sprint performance, vertical jump performance in the combine draft also distinguished between groups, indicating that FG has better performance than EG in SVJ. It suggested that SVJ, which is basketball-specific CMJ in physical fitness [1], was likely to support the slam dunk-related MVJ. Similar to the current study, previous researches in male basketball players competing at national-level, reported that CMJ was strongly correlated with single-leg take-off without dribbling ($P<0.05$, $r = 0.57–0.82$) [11, 15], double-legs take-off without dribbling ($P<0.001$, $r = 0.91$) [15] and single-leg take-off with dribbling ($P<0.05$, $r = 0.51$) [11], which were different types of slam dunk-specific MVJ. However, this significant positive interrelation between CMJ and sport-specific MVJ was also discovered in national-level male volleyball players ($P<0.05$, $r = 0.66–0.89$) [29, 38, 39]. One possible reason refers to that both of SVJ and slam dunk-related MVJ may belong to the slow SSC (contact time>250 ms), which has longer time in eccentric contraction than fast SSC [29]. In this way, slow eccentric contraction during slow SCC, allowed adequate time for force generation, illustrating that players consumed long time on downward phase and transition to

upward phase in SVJ and slam dunk-related MVJ [29]. Due to similar time on force generation, strong association existed in peak positive net joint power(r = 0.92, P<0.01) and positive net work of joint (r = 0.756, P <0.05) between ankle joint of CMJ and lead leg of double-legs MVJ, which may explain the positive correlation between SVJ and slam dunk-related MVJ [14]. Therefore, SVJ may underpin the slam dunk-related MVJ in the contest. For example, although players can spend enough times in approach phase (on the ground) for force generation [29], slam dunk-specific MVJ was a time-limited task, because each players need to finish a slam dunk in limit time during the contest. If FG fails to perform the optimal approach, they still can reach enough flight height for slam dunk by better SVJ performance in the take-off phase. On the contrary, when EG misses the optimal approach, their flight height of slam dunk may significantly decrease. Similar to SVJ, MVJ in the combine draft also differentiated between groups, which suggested that FG possessed higher flight height than EG in MVJ. It suggested that slam dunk-specific MVJ, including single-leg take-off without dribbling [11, 15], double-legs take-off without dribbling [14, 15, 40], single-leg take-off with dribbling [11, 40] and double-legs take-off with dribbling, was likely to support several key explosive tasks in slam dunk contest. One possible reason is that slam dunk-specific MVJ may determine the success rate of alley-opp, which was the typical technique in slam dunk. For example, players in the contest need to reach enough flight height to catch the basketball passed by their partners, or bounced by the ground or backboard in the air within the limit time. Another possible reason refers to that slam dunk-specific MVJ seems to contribute to the difficulty and aesthetics of air-technique. For example, the variety of rim-contacted technique in slam dunk (e.g. hang unilateral or bilateral forearm into the rim, kiss the rim), the number and velocity of air-twist slam dunk (e.g.180 ° twist, 360˚ twist, 720˚ twist), the flight time and height in slam dunk over partner, car or other obstacle, and the flight distance of long distance dunk (e.g.slam dunk from the free-throw line) may all be controlled by slam dunk-specific MVJ. Therefore, in relative to EG, FG was likely to win higher scores during the contest, due to superior performance on slam dunk-specific MVJ. However, except the slam dunk performance, condition after warm-up (sitting vs keepingactively) [41], and foot anthropometry (length of feet and toes) [42] could also probably explain the difference between groups in MVJ. To sum up, TQCS, SVJ and MVJ seem to be key indicators in physical fitness for slam dunk performance.

The application of the study is that, through slam dunk-related anthropometric and physical fitness indicator in this study, the NBA staff can select suitable rookies for slam dunk contest. Similarily, coach from NBA or other basketball league, who want to improve the player's slam dunk performance, should use specific training programs to develop the slam dunk-related indicator. Furthermore, by enhancing on-court slam dunk performance, this study contributes to the audience rating for basketball league, and sports confidence of basketball players [17].

However, this study still has the following limitations. First, although the anthropometric data in NBA combine draft proposes valuable insights, there were certain variations in test protocol and anthropometrist's qualification, which restricted the internal validity of this study. Second, some NBA draft indicators was excluded, due to the large number of missing value [1]. These indicators were not discussed in this study. Besides, the data in this study was anthropometry and physical fitness test in the draft year, which was not the physical condition during slam dunk contest, so sports injury or other disease may occur in the season after draft [1]. However, RSI, exhibiting good reliability on the assessment of jump performance for male basketball players, was not used in the physical fitness test, and it will be used in the future study [16]. Finally, the number of subjects was comparatively low in the study, because only 3–6 rookies were allowed to join the NBA all star slam dunk contest every year [8]. In the future research, members from slam dunk clubs in the world, such as Team Flight Brother

(TFB) in USA, Dunk Elite (DE) in Europe, AirChina (AC) and Dunk King (DK) in China, were recruited as the subjects.

## 5 Conclusions

To conclude, specific anthropometric and physical fitness indicator is clearly different between finals group and elimination group. Height without shoes, standing reach,weight, in anthropometry and three quarter courtsprint, standing vertical jump, max vertical jump in physical fitness are key indicator to slam dunk performance. In accordance with the result in the study, NBA staff can select suitable rookies for slam dunk contest. Similiarly, coach from NBA or other basketball league, who want to improve the player's slam dunk performance, should use specific training programs to develop the slam dunk-related indicator.

## Supporting information

**S1 Data. The raw data used in the study.**
(ZIP)

## Author Contributions

**Conceptualization:** Tse-hau Tong.

**Data curation:** Guo-wei Wang.

**Writing – original draft:** Tse-hau Tong.

**Writing – review & editing:** Tse-hau Tong.

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
