## [Decision Letter · Decision Letter 0]

7 Aug 2023

PONE-D-23-19002Anthropometric and physical fitness indicators in the combine draft between the finalist and the eliminated player in the national basketball association all-star slam dunk contestPLOS ONE

Dear Dr. Tong, 

Thank you for submitting your manuscript to PLOS ONE. After careful consideration, we feel that it has merit but does not fully meet PLOS ONE’s publication criteria as it currently stands. Therefore, we invite you to submit a revised version of the manuscript that addresses the points raised during the review process.

We look forward to receiving your revised manuscript.

Kind regards,

Rafael Franco Soares Oliveira

Academic Editor

PLOS ONE

Journal Requirements:

2. Thank you for stating the following financial disclosure: "No"

3. Thank you for stating the following in your Competing Interests section: "NO authors have competing interests"

6. We note that Figure 2 in your submission contain copyrighted images. All PLOS content is published under the Creative Commons Attribution License (CC BY 4.0), which means that the manuscript, images, and Supporting Information files will be freely available online, and any third party is permitted to access, download, copy, distribute, and use these materials in any way, even commercially, with proper attribution. For more information, see our copyright guidelines: http://journals.plos.org/plosone/s/licenses-and-copyright.

(1) You may seek permission from the original copyright holder of Figure 2 to publish the content specifically under the CC BY 4.0 license. 

(2) If you are unable to obtain permission from the original copyright holder to publish these figures under the CC BY 4.0 license or if the copyright holder’s requirements are incompatible with the CC BY 4.0 license, please either i) remove the figure or ii) supply a replacement figure that complies with the CC BY 4.0 license. Please check copyright information on all replacement figures and update the figure caption with source information. 

If applicable, please specify in the figure caption text when a figure is similar but not identical to the original image and is therefore for illustrative purposes only.

7. We note that Figure 2 includes an image of a participant in the study. 

As per the PLOS ONE policy (http://journals.plos.org/plosone/s/submission-guidelines#loc-human-subjects-research) on papers that include identifying, or potentially identifying, information, the individual(s) or parent(s)/guardian(s) must be informed of the terms of the PLOS open-access (CC-BY) license and provide specific permission for publication of these details under the terms of this license. Please download the Consent Form for Publication in a PLOS Journal (http://journals.plos.org/plosone/s/file?id=8ce6/plos-consent-form-english.pdf). The signed consent form should not be submitted with the manuscript, but should be securely filed in the individual's case notes. 

Please amend the methods section and ethics statement of the manuscript to explicitly state that the patient/participant has provided consent for publication: “The individual in this manuscript has given written informed consent (as outlined in PLOS consent form) to publish these case details”. 

**Additional Editor Comments:**

Dear authors,  I shall say that the manuscript in interesting but the results are quite obvious and make me questioning the novelty of the study. I woud suggest that authors provide a clear explanation on why is this study needed?

Then, 3 reviewers assessed your study. While the reviewer 1 was too positive, reviewers 2 and 3 provided several comments and suggested major revisions. This is also my opinion. 

Beyond the comments made by reviewers, please consider the following: 

In keywords, please avoid using the same words that are already present in the title.

Introduction should provide a clear explanation on why is this study needed? 

The first or second section of material and methods should be the study deseign. Please add it accordingly.

In experimental procedures, invalid/valid indicatiors should be clarified on how were they determined.

In statistical analysis why did authors used Shapiro-Wilk instead of Kolmogorov-Smirnov? Moreover, please clarify why an ANOVA was applied? It seems that independent sample T-test would be enough. Nonetheless, the results section should be revised to avoid repetition (both text and tables present identical information) and to provide more inferential statistics.  

Furthermore, discussion should be revised because six pages are too much. 

Please check the reference format in the final list. It is not following guidelines of Plos One. Best regards

Reviewers' comments:

Reviewer's Responses to Questions

**Comments to the Author**

1. Is the manuscript technically sound, and do the data support the conclusions?

Reviewer #1: Yes

Reviewer #2: Partly

Reviewer #3: Yes

2. Has the statistical analysis been performed appropriately and rigorously? 

Reviewer #1: Yes

Reviewer #2: I Don't Know

Reviewer #3: Yes

3. Have the authors made all data underlying the findings in their manuscript fully available?

Reviewer #1: Yes

Reviewer #2: Yes

Reviewer #3: Yes

4. Is the manuscript presented in an intelligible fashion and written in standard English?

Reviewer #1: Yes

Reviewer #2: Yes

Reviewer #3: Yes

5. Review Comments to the Author

Reviewer #1: First of all, the reviewer would like to thank the authors for their work and efforts in trying to improve sports science knowledge. Overall, the study is well designed and well-written, with a great original article evaluating the usefulness of the topic.

Accepted without revision

Reviewer #2: Thank you for your submitted manuscript entitled, “Anthropometric and physical fitness indicators in the combine draft between the finalist and the eliminated player in the national basketball association all-star slam dunk contest’’. The manuscript is interesting and well written. However, I have a few comments that I suggest you consider before publishing the text. In this reviewer’s opinion, there are major issues that limit the scientific relevance of the present study. The concern is that the number of subjects is relatively low.

ABSTRACT

• ‘’No study reported anthropometric and physical indicator for competitors of dunk contest’’. Pease clarify the statement

• Why not use average of age, height and body mass index of players.

• Why not use average (Mean and SD) of the physical test measurement in the abstract

• Could be a relevant conclusion of the present study to find what is important to know.

INTRODUCTION

• The development of the introduction needs to be more hypotheses driven and develop the questions leading up to the section in the methods section.

• The introduction is consistent and easy to follow. Hypotheses are clearly formulated.

• The Authors should clarify the actual heritage of this study. I am concerned about the originality of the present study.

METHOD

• How was sample size determined? (Sampling technique!)

• Ethical Consideration: Make sure you have the proper informed consent statement in the paper, i.e., subjects were informed of the experimental risks and signed an informed consent document prior to the investigation.

STATISTICAL ANALYSIS

• Please, present methods of data analysis and criterion of results interpretation.

• Please define specificity and sensitivity in this research context

RESULTS

• Results description is a little chaotic and insufficient.

• Please, add some introductions to the description of the results and indicate what and why you did. Each result presented in the tables should be commented on in the text.

DISCUSSION

• The discussion needs to reflect what you found, how it relates to the literature and then what it means physiologically or from a practical aspect and each paragraph should be logical in sequence as at present it is a bit hard to follow.

• Make sure the paper’s importance and need is clear to the reader.

CONCLUSION

• Thus, my biggest concern is that the practical and cases of this particular study are not particularly high in the coach after reading this is not really thinking about changing any of their approaches to training or testing. Our main goal was not to provide changing of approaches to training or testing.

• Why might one want to cite this paper? What is the true impact of the literature?

Reviewer #3: General Comments

The manuscript addresses an interesting topic, presents and discusses data from a perspective not yet addressed, and is of potential interest to basketball professionals and researchers.

Nevertheless, to be publishable, the authors should make improvements mainly in the introduction and methodology, adding missing information and clarifying some information already presented.

The limitations of the study, both internally and externally, are considerable. However, the authors present them in the discussion. Considering the use of already available data (and not collected specifically for the study), the limitations are understandable.

Abstract

The penultimate sentence of the abstract is a bit confusing and long. Please try to make it more direct and easily understandable.

Introduction

Page 9, line 4 – the term "contained" seems to me not to be the most appropriate, please review.

Page 9, lines 18-19 – The 1st sentence of this 2nd paragraph is confusing. Please specify clearly the audience referred to and the sporting season to which it refers. Or does this audience refer to all seasons between 2013-2018?

The authors talk about the "draft" and "slam dunk contest" as if the reader is already fully familiar with this selection and competition process. Given that this manuscript may be of interest to a wider variety of readers, including colleagues not as familiar with these NBA events, it is advisable to introduce them in the introduction.

Page 10, line 7 – the authors refer to reference number 48, however the list of references only goes up to number 46. Please check.

Page 10, lines 2-8 – The authors present anthropometric characteristics and references that support slam dunk performance in basketball players, however some of these references refer to the sport of volleyball (ref 32 and 43), another to athletes in general (ref 34) and another even more general reference (45). Please reconsider if all the references in this sentence are well selected.

Page 10, lines 14-16 – The hypothesis regarding the anthropometric component is misplaced. The authors state that they expect "that finalist has greater value in the anthropometry", please clarify which anthropometric variables you expect to find this "greater value" in. Anthropometry is the study of the physical measurements and dimensions of the human body, not "the measurements themselves". By saying "greater value in the anthropometry" the reader can interpret for example greater body mass, or greater amount of fat mass, which does not seem to me to be the hypothesis raised.

Materials and methods

Table 1 – The last variable presented is "Score Mean (point)", the legend of this variable is unclear, are we talking about average points during the competition? Average of the various competitions? Please clarify.

Page 11, Participants – The authors say the sample consisted of 49 participants. However, due to the criteria they present, they had to eliminate 17, so the final sample and, subject to statistical treatment was only 32 athletes. In other words, the sample of the present study is 32 athletes and not 49, so this information should be clarified both in the "sample" section and in the abstract.

Page 12, Antropometrics Indicators – Information regarding the assessment of anthropometric characteristics is insufficient. The authors report that measurements were performed "by the NBA conditioning coach by valid and reliable protocol". Can you ensure that these coaches were certified anthropometrists? If they were not, these measurements are no longer scientifically valid, as there was no control over measurement error of anthropometric variables. Please clarify this topic. What anthropometric protocols were used? A brief description that does not fully clarify the protocols used is presented in Table 2. For example, the description given for the variable "height" does not mention the positioning of the head. According to the ISAK (International Society for the Advancement of Kinathropometry) protocol, the head would be positioned in the Frankfurt plane. Please complement and support with references, which anthropometric protocols are used.

Page 13, Statistical Analysis – The authors applied one-way ANOVA, noting that both groups had normal distribution of the data. Another assumption for the application of this statistical test is the homegeneity of variables. Was this assumption tested?

Page 14, Results, lines 4-5 – Consider replacing the phrase "was significantly less than" by "had significantly lower values than".

Table 2 – Please add the acronym of each test to the table, it can help the reader to “incorporate” the acronym.

Discussion

Page 15, lines 3-4 – Please reconsider the presentation of the hypothesis regarding anthropometry data, in line with the previous suggestion.

Page 15, line 19 – Please replace “Figure II” by “Figure 2.”. Is the image not copyrighted? Do you have permission to use it?

Page 15, lines 18-22 – Not all readers may know that aesthetic is accounting in slam dunk. It is really important that this information is clear in the introduction and can be reinforced when presented in the discussion.

Page 16, lines 1-2 – Please clarify how a shorter limb (arm or lower extremity) can produce a higher greater velocity. In this actual form, it seems that you are saying that a shorter arm produces a higher angular velocity. I supposed that isn´t it that you want to say, maybe you are talking about the tendon insertion?

Page 16, line 13 – Please replace “dead mass” by fat mass, we all hope that participant do not have dead mass.

Page 18, line 1 – Please linguistically revise this 1st line.

6. PLOS authors have the option to publish the peer review history of their article (what does this mean?). If published, this will include your full peer review and any attached files.

Reviewer #1: No

Reviewer #2: **Yes: **Souhail Hermassi

Reviewer #3: No

---

## [Author Response · Author response to Decision Letter 0]

5 Oct 2023

Dear Editor and Reviewers:

Thank you for your letter and for the reviewers’ comments concerning our manuscript entitled “Anthropometric and physical fitness indicators in the combine draft between the finalist and the eliminated player in the national basketball association all-star slam dunk contest” (Manuscript Number: PONE-D-23-19002). Those comments are all valuable and very helpful for revising and improving our paper, as well as the important guiding significance to our research. We have studied comments carefully and have made corrections which we hope to meet with approval. We highlighted all revision in red color. The corrections in the paper and the responds to the referee’s comments are as following:

Additional Editor Comments：

Comment 1. I shall say that the manuscript in interesting, but the results are quite obvious and make me questioning the novelty of the study. I would suggest that authors provide a clear explanation on why is this study needed?

OUR REPLY: Thank for your encouragement and suggestion for this study. Based on your question, we made the corresponding explanation as follows:

1. The novelty of this study: The slam dunk-related indicator in the anthropometry (height without shoes, standing reach, and weight) and physical fitness (three quarter court sprint, standing vertical jump, and max vertical jump) was first discovered in this study. However, the interesting result, which is opposite to previous studies, was found in our study. At first, we found that the NBA players with good slam dunk performance has shorter height without shoes and standing reach, less weight than eliminated players, while relevant studies concluded that taller player with higher standing reach has better basketball-related performance. Previous viewpoints considered that the distance between these players and the rim is shorter. Second, our result revealed that finalist’s weight was lighter than their peers. In addition, it also opposite to previous conclusion, reporting that basketball player with larger weight and less body fat percentage has better on-court performance. Third, this study was concluded that standing vertical jump is most critical indicator in slam dunk-related physical fitness. In addition, it is also different from empirical viewpoint among basketball coaches. They believed that max vertical jump is vital to slam dunk, because it decides the success rate and difficulty of flight technique. To sum up, several interesting results and new slam dunk-related indicators indicated the novelty of this study.

2. The importance of our study: Based on your question, we explained that the study has practical function and academic impact in the field of sports. as follows:

(1) The current research has several functions in practical aspects. First, according to the slam dunk-related indicator in this study, staff from NBA or other basketball league can select suitable rookies to improve the aesthetic and quality of the slam dunk contest. For example, they can invite the basketball players into dunk contest, who has short height without shoes and standing reach, light weight with good performance on three quarter sprint, standing vertical jump and max vertical jump. Second, NBA coaches or coaches from other basketball league, who want to improve the player’s slam dunk performance, will use specific training programs, such as plyometric training (PT) with or without weight, velocity-based training (VBT), post-activation potentiation (PAP), aiming to develop the slam dunk-related indicator after reading this study. On the contrary, if coaches arrange the specific training program for slam dunk performance, only based on their experience or studies about other basketball-related performance, they will discover many possible influence factors for slam dunk, including flexibility, postural stability, speed of approach and take-off, dribbling technique, flight height and distance. Third, good slam dunk performance will contribute to the audience rating of basketball league. Fourth, because slam dunk is aesthetic and ornamental, one successful on-court slam dunk will improve the sports confidence of basketball player by promotion of functionality appreciation. One recent research reported the positive correlation between functionality appreciation and sports confidence among athletes (p<0.036) (Ricketts C, Malete L, Myers ND, Bateman AG, Bateman CJ. Sport bodies: An examination of positive body image, sport-confidence, and subjective sport performance in Jamaican athletes. Psychology of Sport & Exercise. 2023; 67: 102434).

(2) From the academic aspect, our study will provide relevant evidence and attract more researchers to investigate slam dunk in basketball. First, strength and conditioning researchers will show interest in this study, because they can discover more slam dunk-related indicators for different type of subjects (national-level basketball athlete, state-level basketball players, collegiate-level players, and slam dunk players from slam dunk clubs) according to our study. Another reason is that they will able to design the specific training program for slam dunk performance, because our study indicates the key physical fitness indicators. Second, researchers in basketball training will cite the study, because based on the result of this study, they can explore more key physical fitness indicators, which is correlated with different tasks of on-court slam dunk, including slam dunk with or without defense, slam dunk with dribbling, alley-oop (dunk without dribbling). Third, researchers in the field of sports biomechanics may read our study. The reason is that there was potential new mechanism of biomechanics (kinematics: different joint angle and angular velocity, kinetics: different vertical ground reaction force, human material mechanics: different vertical stiffness, joint stiffness, stress and strain) in slam dunk, which is a complex technique combination in different situation. It attracts them to analyze the motion of slam dunk. 

(3) To sum up, several practical function and possible academic impact that we presented, is the reason of why this study need. 

3. We also made corresponding revisions. as follows:

 Introduction section:

“Thus, research assessing the basketball players, needs to propose the novel insights about players in the slam dunk contest to define their own characteristic in anthropometry and physical fitness. These insights allow the basketball coaches from NBA or other basketball league to develop the slam dunk performance of players, based on the slam dunk-related physical fitness, contributing to player’s sports confidence [35] and basketball league’s audience rating. Identification of potential differences on anthropometric and physical fitness indicator between different performances in the slam dunk contest (finalist vs eliminated player) can inform the slam dunk-related indicator for selecting the suitable participants into the contest, which can maximize its aesthetic [18] and audience rating.”

(2) “application of the study” section in Discussion section: 

“The application of the study is that, through slam dunk-related anthropometric and physical fitness indicator in this study, the NBA staff can select suitable rookies for slam dunk contest. Similarily, coach from NBA or other basketball league, who want to improve the player’s slam dunk performance, should use specific training programs to develop the slam dunk-related indicator. Furthermore, by enhacing on-court slam dunk performance, this study contributes to the audience rating for basketball league, and sports confidence of basketball players [35].”

(3) Conclusion section: 

“To conclude, specific anthropometric and physical fitness indicator is clearly different between finals group and elimination group. Height without shoes, standing reach, weight, in anthropometry and three quarter court sprint, standing vertical jump, max vertical jump in physical fitness are key indicator to slam dunk performance. In accordance with the result in the study, NBA staff can select suitable rookies for slam dunk contest. Similiarly, coach from NBA or other basketball league, who want to improve the player’s slam dunk performance, should use specific training programs to develop the slam dunk-related indicator.”

Comment 2. In keywords, please avoid using the same words that are already present in the title.

OUR REPLY: We appreciate it very much for this good suggestion. According to your valuable suggestion, we changed the “Anthropometry; Basketball; Physical fitness” to “Somatotype; Sports Performance; Agility; Sprinting; Vertical Jump” in keywords as follows:

Keywords: Somatotype; Sports Performance; Agility; Sprinting; Vertical Jump.

Comment 3. Introduction should provide a clear explanation on why is this study needed?

OUR REPLY: Thank for your helpful suggestions. We presented a clear explanation of that why we study the anthropometric and physical fitness indicators of players in the slam dunk contest and that why we identify the potential differences of these indicators between different performances in the contest (finalist vs eliminated player) in Introduction section as follows:

“Therefore, competitors with different sports performance in the NBA slam dunk contest, were likely to own special characteristic on anthropometry and physical fitness. Finalist in the contest probably possesses optimal body fat percentage [3-4], height and standing reach [3-4, 16, 24, 36], weight [3-4, 16, 24, 36],wingspan [3-4] in the anthropometry, and extremely outstanding performance on linear sprint [3-4, 16, 37], standing vertical jump and countermovement jump [24, 27, 29], max vertical jump [3-4, 24, 27, 29], and reactive strength index (RSI) [33] in the physical fitness test, supporting their slam dunk-specific performance. By contrast, some players were eliminated in the preliminary, because of poor slam dunk performance, which may be resulted from unsuitable anthropometric characteristic and unsatisfactory physical fitness.”

“Thus, research assessing the basketball players, needs to propose the novel insights about players in the slam dunk contest to define their own characteristic in anthropometry and physical fitness. These insights allow the basketball coaches from NBA or other basketball league to develop the slam dunk performance of players, based on the slam dunk-related physical fitness, contributing to player’s sports confidence [35] and basketball league’s audience rating. Identification of potential differences on anthropometric and physical fitness indicator between different performances in the slam dunk contest (finalist vs eliminated player) can inform the slam dunk-related indicator for selecting the suitable participants into the contest, which can maximize its aesthetic [18] and audience rating.”

Comment 4. The first or second section of material and methods should be the study design. Please add it accordingly.

OUR REPLY: We appreciate it very much for this good suggestion. We have searched the studies using the same research method, only analyzing the stats from NBA official database in the study. The research that we refer to is as follows:

1. Zhang S, Lorenzo A, Gómez MA, Mateus N, Gonçalves B, Sampaio J. Clustering performances in the NBA according to players' anthropometric attributesand playing experience. Journal of Sports Sciences.2018; 36(22): 2511-2520.

2. Igor R, Radivoj M, Marko C, Predrag B, Milivoj D. NBA Pre-Draft Combine is the weak predictor of rookie basketball player’s performance. Journal of Human Sport and Exercise.2020; 16(3):1-10.

Based on your suggestions, we have added “study design” in the first section of Material and Methods section as follows:

2.1 Study design

“The descriptive design was used [36]. Available data from NBA official combine draft database from 2000 to 2015 was analyzed in this study (available at www.nba.com/stats/draft/combine) [8-10]. The raw data that this study used is available as supplementary material S1 Data.”

Comment 5. In experimental procedures, invalid/valid indicators should be clarified on how were they determined.

OUR REPLY: Thank for your helpful suggestion. The original plan of this study is that all indicators in the anthropometry and physical fitness were include as valid indicator in the study, because we believed that all indicators may be related to slam dunk performance. Unfortunately, if all indicators used are used as valid indicator in the study, the number of the subjects is minimal, because some indicators have excessive missing rate. Thus, we developed the criterion of invalid/valid indicator to maintain the number of subjects. 

On the basis of your suggestions, we have clarified on how invalid/valid indicator was determined in “experimental procedures” section of Materials and Methods section as follows:

“The experimental procedure in this study was presented in the Fig 1. At first, the invalid indicator was excluded. Criterion for invalid indicator was or extremely correlated with other indicators (r >0.9) and excessive missing rate (missing rate: >50%), indicating the ratio of how many subjects missing the data of this indicator [4]. According to the criterion, height with shoes (r >0.9), which extremely interrelate with height without shoes in previous study [4], hand length (missing rate: >53.06%), hand width (missing rate: >53.06%), in the anthropometry as well as shuttle run (missing rate: >61.22%), max bench press (missing rate: >90%) in the physical fitness test was excluded [8-10]. The reason of excluding the height with shoes rather than height without shoes was that each NBA players owned their unique trainers and shoe pods with various thickness. This may result in the deviation of measurement. Then, the valid indicator was included. Criterion for valid indicator was that indicator has low missing rate (missing rate: ＜50%) and not extremely interrelated with other indicators (r <0.9). Based on this criterion, body fat percentage (BFP), height without shoes (height W/O shoes), standing reach (SR), weight, wingspan, lane agility time (LAT),three quarter court sprint (TQCS),standing vertical jump (SVJ), and max vertical jump (MVJ) were included as valid indicator in the study (missing rate: 0%, r <0.9).”

Comment 6. In statistical analysis why did authors used Shapiro-Wilk instead of Kolmogorov-Smirnov? 

OUR REPLY: Thank you very much for pointing out. We have made the corresponding explanation as follows:

One study reported the relevant evidence that when sample size is from 10 to 2000 (N=10-2000), Shapiro-Wilk is the most powerful normality distribution test, while Kolmogorov-Smirnov is the least power test (Razali NM, Wah YB. Power comparisonsof Shapiro-Wilk, Kolmogorov-Smirnov, Lilliefors and Anderson-Darling tests. Journal of Statistical Modeling and Analytics.2011; 2(1): 21-33). 

Thus, when sample size is between 10 and 2000 (N=10-2000), the Shapiro-Wilk test is performed. In this study, the Shapiro-Wilk test instead of Kolmogorov-Smirnov was used, because the sample size to statistical analysis is 36 (N=10-2000). In addition, it indicated that the power of Shapiro-Wilk test is larger than Kolmogorov-Smirnov test in this range of sample size.

Comment 7. Moreover, please clarify why an ANOVA was applied? It seems that independent sample T-test would be enough.

OUR REPLY: Thank you very much for pointing out. When we explored the intergroup difference between FG and EG, we realized that both of one-way ANOVA and independent sample t-test indicated the same significance level. 

Thus, based on your suggestion, we employed the independent sample t-test instead of one-way ANOVA as statistical analysis and removed the partial eta square, the effect size of one-way ANOVA as follows:

(1) “statistical analysis” section in Materials and Methods section:

“At first, descriptive statistics was adopted for all the data in the study. It was presented as mean and standard deviation (Mean ± SD). Next, Shapiro–Wilks analysis was adopted for testing the normal distribution of data. Levene test was used to examine the homogeneity of variables. The independent sample t-test was employed to test the statistical significance of intergroup difference, while the data met the normal distribution and the homogeneity of variables.”

 “Cohen’s d level was employed as effect size to illustrate the extent of intergroup difference and interpreted as: cohen’s d<0.3, trivial; 0.3≤cohen’s d<0.7, small; 0.7≤cohen’s d<1.3, moderate; 1.3≤cohen’s d<2.1, large; 2.1≤Cohen’s d<4.0, very large; cohen’s d≥4.0, extreme large [13-14]. The significance level of normal distribution and homogeneity of variables was set at P＞0.05. In addition, the statistical significance of independent sample t-test was set at P＜0.05. The statistical analysis was performed using SPSS Statistics 27.0 (IBM Corp, Armonk, New York, USA). ”

 “method” section in Abstract section:

“Independent sample t-test with cohen’s d was adopted for evaluating the statistical significance of intergroup difference and its effect size.”

Comment 8. Nonetheless, the results section should be revised to avoid repetition (both text and tables present identical information) and to provide more inferential statistics. 

OUR REPLY: We appreciate it very much for this good suggestion. Based on your suggestion, we made the corresponding revision for avoiding repetition and providing more inferential statistics in Result section as follows:

“The descriptive statistics for anthropometric indicators was displayed in Table 4. Independent sample t-test suggested that FG had significantly lower values than EG onheight W/O shoes, SR, and weight (P＜0.05), while the intergroup difference on BFP and wingspan were not of statistical significance (P>0.05) (see Table 4). ES test about cohen’s d demonstrated that extent of intergroup difference was moderate on height W/O shoes, SR, and weight (0.7≤cohen’s d<1.3), while it was small on BFP and wingspan (0.3≤cohen’s d<0.7) (see Table 5).”

“The descriptive statistics for physical fitness indicators was shown in Table 4. The result of independent sample t-test showed that FG has significant better performance than EG on TQCS, SVJ, and MVJ (P＜0.05), while there was no significant intergroup difference on LAT (P>0.05) (see Table 4). ES test about cohen’s d suggested that extent of intergroup difference was moderate on TQCS, SVJ and MVJ (0.7≤cohen’s d<1.3), whereas it was small on LAT (0.3≤cohen’s d<0.7) (see Table 5).”

Comment 9. Furthermore, discussion should be revised because six pages are too much.

OUR REPLY: We appreciate it very much for this good suggestion. We tried our best to simplify the Discussion section and to remove the irrelevant content. At first, we have rewritten the Discussion section. Next, we have deleted the irrelevant reference. The deleted reference is presented as follows:

1.Torres-unda J, Zarrazquin I, Gil J, Ruiz F, Irazusta A, Kortajarena M, et al.Anthropometric, physiological and maturational characteristics in selected elite and non-elite male adolescent basketball players. Journal of Sports Sciences. 2013; 31(2): 196-203.

2. Mtsweni LB, West SJ, Taliep MS. Anthropometric and physical fitness characteristics of female basketball players in South Africa. South African Journal for Research in Sport Physical Education & Recreation. 2017; 39(3): 93-103.

3. Ramos S, Volossovitch A, Ferreira AP, Fragos I, Massuca LM. Training experience and maturational, morphological, and fitness attributes as individual performance predictors in male and female under-14 Portuguese elite basketball players. Journal of Strength & Conditioning Research. 2021; 35(7): 2025-2032.

4. Spaniol FJ. Striking skills: Developing power to turn. Strength Conditioning Journal. 2012; 34(6): 57-60.

5. Waldron M, Worsfold P, Twist C, Lamb K. Changes in anthropometry and performance, and their interrelationships, across three seasons in elite youth rugby league players. Journal of Strength & Conditioning Research. 2014; 28(11): 3128-3136.

6. Amonette WE, Brown D, Dupler TL, XU JH, Tufano JJ, DeWitt JK. Physical determinants of interval sprint times in youth soccer players. Journal of Human Kinetics. 2014; 40: 113-120.

7. DeAlmeida-neto PF, DeMatos DG, Baxter-jones ADG, Batista GR, Pinto VCM, Dantas M, et al. The effectiveness of biological maturation and lean mass in relation to muscle strength performance in elite young athletes. Sustainability.2020; 12(17):6696.

8. Mehran N, Williams PN, Keller RA, Khalil LS, Lombardo SJ, Kharrazi FD.Athletic performance at the national basketball association combine after anterior cruciate ligament reconstruction. Orthopaedic Journal of Sports Medicine. 2016; 4(5): 2325967116648083.

9. Pamuk Ö, Makarac Y, Ceylan L, Küçük H, Kızılet T, Ceylan T, et al.Associations between force-time related single-leg counter movement jump variables, agility, and linear sprint in competitive youth male basketball players. Children.2023; 10(3): 427.

10. Struzik A, Winiarski S, Popowczak M, Rokita A. Relationships between variables describing vertical jump and sprint time. South African Journal for Research in Sport Physical Education and Recreation. 2017; 39(1): 177-88.

11. Koklu Y, Alemdaroglu U, Ozkan A, Koz M, Ersoz G. The relationship between sprint ability, agility and vertical jump performance in young soccer players. Science & Sport. 2015; 30(1): E1-E5.

12. Kipp K, Kiely M, Giordanelli M, Malloy P, Geiser C. Joint- and subject-specific strategies in male basketball players across a range of countermovement jump heights. Journal of Sports Sciences. 2020; 38(6): 652-657.

Comment 10. Please check the reference format in the final list. It is not following guidelines of Plos One.

OUR REPLY: Thank for your helpful suggestions. Based on your suggestions, we revised the reference to meet the guidelines of Plos One. 

Reviewer#1:

Comment 1. First of all, the reviewer would like to thank the authors for their work and efforts in trying to improve sports science knowledge. Overall, the study is well designed and well-written, with a great original article evaluating the usefulness of the topic.

OUR REPLY: Thank for your admiration and encouragement on this study. It stimulates all authors of the study to explore new knowledge from this topic in the future. Although evidence about this topic (slam dunk performance in basketball) is seldom from various sources, resulting in different levels of difficulty, we will make great efforts to improve this study during the revision process.

Reviewer#2:

Comment 1. Thank you for your submitted manuscript entitled, “Anthropometric and physical fitness indicators in the combine draft between the finalist and the eliminated player in the national basketball association all-star slam dunk contest’’. The manuscript is interesting and well written. However, I have a few comments that I suggest you consider before publishing the text. In this reviewer’s opinion, there are major issues that limit the scientific relevance of the present study. The concern is that the number of subjects is relatively low.

OUR REPLY: Thank for your encouragement and suggestion for the study. According to the number of subjects in our study, we have made the corresponding explanations as follows:

The possible reason is that the number of NBA players, who participated in the all star slam dunk contest is comparatively low, because only 3-6 rookies was allowed to join the contest every year. In addition, the data in this study came from the available database (www.nba.com/stats/draft/combine), and thus we were unable to add more subjects to this study. In future study, we try to recruit the members of slam dunk clubs in the world, such as Team Flight Brother (TFB) in USA, Dunk Elite (DE) in Europe, AirChina (AC) and Dunk King (DK) in China, because they have excellent sports performance in slam dunk contest. Thus, the anthropometric parameters and physical fitness indicator of them may be more typical for the research. According to your valuable suggestion, we made the corresponding revision in “limitation of the study” section in Discussion section as follows:

“However, this study still has the following limitations. First, some NBA draft indicators was excluded, due to the large number of missing value [4]. These indicators were not discussed in this study. Second, the data in this study was anthropometry and physical fitness test in the draft year, which was not the physical condition during slam dunk contest. Besides, sports injury or other disease may occur in the season after draft [4]. However, RSI, exhibiting good reliability on the assessment of jump performance for male basketball players, was not used in the physical fitness test, and it will be used in the future study [33]. Finally, the number of subjects was comparatively low in the study, because only 3-6 rookies were allowed to join the NBA all star slam dunk contest every year [34]. In the future research, members from slam dunk clubs in the world, such as Team Flight Brother (TFB) in USA, Dunk Elite (DE) in Europe, AirChina (AC) and Dunk King (DK) in China, were recruited as the subjects.”

Comment 2. ABSTRACT: No study reported anthropometric and physical indicator for competitors of dunk contest’’. Please clarify the statement.

OUR REPLY: We appreciate it very much for this good suggestion. Based on your constructive suggestion, we make the corresponding explanations and revision: 

We want to clarify that in the published study, no study discussed the anthropometric and physical fitness data in combine draft from participants of NBA all star slam dunk contest. We have revised this sentence as follows:

“Little is known about the difference of anthropometry and physical fitness between the finalist and eliminated player in the NBA all star slam dunk contest.”

Comment 3. ABSTRACT: Why not use average of age, height and body mass index of players.

OUR REPLY: Thank you very much for pointing out. Based on your question, we have made the following explanation and revision:

1. The reason of why we not use average of age as indicator is as follows: To the best of our knowledge, by searching the date of birth, draft year and competition year of the subjects, we can acquire two indicators concerning age included age in draft year and age in participating NBA all star slam dunk contest. Due to the lack of data during slam dunk contest, it becomes extremely difficult to find the correlation between the age and slam dunk performance in this study. 

2. Body Mass Index (BMI) is a general indicator which can be used to assess the health condition of human, but it is not appropriate for anthropometry of NBA players. The reason is that in this study, NBA players possess large weight (79.1 kg-126.37 kg) and low body fat percentage (BFP) (4.1%- 10.5%), indicating that their lean body mass (LBM) is significantly larger than ordinary subject or non-athlete. In the meanwhile, BMI is often affected by the weight, but not reflects the key indicator of body composition (BFP, LBM) in NBA players. As a result, for the anthropometric assessment of NBA players, BFP is more suitable than BMI.

3. We selected the height without shoes, instead of height, as indicator. One possible reason refers to that each NBA players owned their unique trainers and shoe pods with various thickness, it may result in the deviation of measurement, if we were unable to control this confounding factor. Therefore, in relative to the height, the height without shoes is a better indicator for reflecting the actual height of NBA players.

4. According to your suggestion and above reasons, we have supplemented the average of age in draft year, height without shoes and body fat percentage in “method” section of Abstract section and “participants” section of “Materials and Methods” section as participants’ characteristics:

 “method” section of Abstract section: 

“Draft data of 32 basketball players (N=32, age in draft year: 20.69±2.28 years old, height without shoes: 196.75±8.68 cm, weight: 96.85±10 kg, body fat percentage: 6.07±1.23%) participating in the 2000-2015 draft and 2003-2023 slam dunk contest was selected from national basketball association database. It was classified into finals group (FG) (N=16) and elimination group (EG) (N=16).”

(2) “participants” section in the Materials and Methods section:

“The combine draft data of 32 NBA players from NBA official draft database was sampled as participants in the present study（N=32, age in draft year: 20.69±2.28 years old, height without shoes: 196.75±8.68 cm, weight: 96.85±10 kg, body fat percentage: 6.07±1.23%）[8-10].Table 1 displays participants’ characteristics.”

Comment 4. ABSTRACT: why not use average (Mean and SD) of the physical test measurement in the abstract.

OUR REPLY: Thank you for your helpful comments. On the basis of your important suggestion, we added the average (Mean and SD) in the Abstract section as follows:

“The result indicates that Finalist group was significant less than elimination group on height without shoes (FG vs EG: 193.43±9.47 cm vs 200.06±6.52 cm, P＜0.05), standing reach (FG vs EG: 257.66±12.32 cm vs 268.29±10.03 cm, P＜0.05) and weight (FG vs EG: 93.38±7.37 kg vs 100.33±11.25 kg, P＜0.05). Conversely, compared to elimination group, finalist group has significant better performance on three quarter court sprint (FG vs EG: 3.15±0.1 s vs 3.26±0.12 s, P＜0.05) ,standing vertical jump (FG vs EG: 84.88±5.13 cm vs 78.83±4.9 cm, P＜0.05) and max vertical jump (FG vs EG: 102.39±6.47 cm vs 94.79±8.34 cm, P＜0.05).”

Comment 5. ABSTRACT: Could be a relevant conclusion of the present study to find what is important to know.

OUR REPLY: Thank you very much for pointing out. We make corresponding explanation:

1. At first, we discovered key indicator (anthropometry: height with shoes, standing reach, weight in anthropometry and three-quarter court sprint, standing vertical jump, max vertical jump in physical fitness) of slam dunk performance. It provided suggestion to select the potential participants for slam dunk contest and to improve the slam dunk performance.

2. Second, the study could increase the audience rating of basketball league and sports confidence of basketball players through enhancing the aesthetic of on-court slam dunk. One recent research illustrated the relevance between aesthetics of slam dunk and sports confidence (Ricketts C, Malete L, Myers ND, Bateman AG, Bateman CJ. Sport bodies: An examination of positive body image, sport-confidence, and subjective sport performance in Jamaican athletes. Psychology of Sport & Exercise. 2023; 67: 102434). The result of RICKETTS et al. suggested that functionality appreciation (including aesthetic of slam dunk) was significantly correlated with sports confidence (p<0.036).

3. However, we made corresponding revisions as follows:

 “conclusion” section of Abstract section:

“To conclude, specific anthropometric and physical fitness indicator shows clear difference between finals group and elimination group. Height without shoes, standing reach, weight in anthropometry and three quarter court sprint, standing vertical jump, and max vertical jump in physical fitness are key indicator to slam dunk performance. In line with the result in the study, NBA staff can select suitable rookies for slam dunk contest. Similiarly, coach from NBA or other basketball league, who want to improve the player’s slam dunk performance, should use specific training programs to develop the slam dunk-related indicator.”

 Conclusions section:

“To conclude, specific anthropometric and physical fitness indicator is clearly different between finals group and elimination group. Height without shoes, standing reach, weight, in anthropometry and three quarter court sprint, standing vertical jump, max vertical jump in physical fitness are key indicator to slam dunk performance. In accordance with the result in the study, NBA staff can select suitable rookies for slam dunk contest. Similiarly, coach from NBA or other basketball league, who want to improve the player’s slam dunk performance, should use specific training programs to develop the slam dunk-related indicator.”

 “application of the study” section in Discussion section:

“The application of the study is that, through slam dunk-related anthropometric and physical fitness indicator in this study, the NBA staff can select suitable rookies for slam dunk contest. Similarily, coach from NBA or other basketball league, who want to improve the player’s slam dunk performance, should use specific training programs to develop the slam dunk-related indicator. Furthermore, by enhancing on-court slam dunk performance, this study contributes to the audience rating for basketball league, and sports confidence of basketball players [35].”

 Introduction section:

“Thus, research assessing the basketball players, needs to propose the novel insights about players in the slam dunk contest to define their own characteristic in anthropometry and physical fitness. These insights allow the basketball coaches from NBA or other basketball league to develop the slam dunk performance of players, based on the slam dunk-related physical fitness, contributing to player’s sports confidence [35] and basketball league’s audience rating. Identification of potential differences on anthropometric and physical fitness indicator between different performances in the slam dunk contest (finalist vs eliminated player) can inform the slam dunk-related indicator for selecting the suitable participants into the contest, which can maximize its aesthetic [18] and audience rating.”

Comment 6. INTRODUCTION: The development of the introduction needs to be more hypotheses driven and develop the questions leading up to the section in the methods section.

OUR REPLY: Thank you for your helpful comments. According to your suggestion, we made corresponding revision in Introduction section as follows: 

1. At first, we have revised the Introduction section to clarify why we discuss anthropometric and physical fitness characteristic for participants in slam dunk contest, and why we identify the difference on anthropometry and physical fitness between finalist and eliminated player (hypothesis in the study). It made the Introduction section more hypotheses driven and proposed the questions leading up to the section in the methods section. Our revision is presented as follows:

“Thus, research assessing the basketball players, needs to propose the novel insights about players in the slam dunk contest to define their own characteristic in anthropometry and physical fitness. These insights allow the basketball coaches from NBA or other basketball league to develop the slam dunk performance of players, based on the slam dunk-related physical fitness, contributing to player’s sports confidence [35] and basketball league’s audience rating. Identification of potential differences on anthropometric and physical fitness indicator between different performances in the slam dunk contest (finalist vs eliminated player) can inform the slam dunk-related indicator for selecting the suitable participants into the contest, which can maximize its aesthetic [18] and audience rating.”

2. Next, we presented the potential differences on anthropometric and physical fitness indicator between different performances in the slam dunk contest (finalist vs eliminated player). It explained why we propose this hypothesis. Our revision is presented as follows:

“Therefore, competitors with different sports performance in the NBA slam dunk contest, were likely to own special characteristic on anthropometry and physical fitness. Finalist in the contest probably possesses optimal body fat percentage [3-4], height and standing reach [3-4, 16, 24, 36], weight [3-4, 16, 24, 36],wingspan [3-4] in the anthropometry, and extremely outstanding performance on linear sprint [3-4, 16, 37], standing vertical jump and countermovement jump [24, 27, 29], max vertical jump [3-4, 24, 27, 29], and reactive strength index (RSI) [33] in the physical fitness test, supporting their slam dunk-specific performance. By contrast, some players were eliminated in the preliminary, because of poor slam dunk performance, which may be resulted from unsuitable anthropometric characteristic and unsatisfactory physical fitness.”

3. Finally, we added the introduction of slam dunk contest and the distinction of sports performance between NBA slam dunk contest and NBA regular season in Introduction section for hypotheses driven as follows:

“Though all the rookies with less than three year of NBA experience were allowed to apply for the contest, the number of competitors was kept between three and six [34]. All the competitors drew lots to decide their order of play before the preliminary. After one player completed a slam dunk in limit time, each referee should give one individual score from six to ten, while the sum of all referee’s score is the total score of one slam dunk. Each referee can give six as individual score, if one competitor failed to make one slam dunk in required time. Competitors, who ranked in top two or top three in preliminary, can reach the finals. When both of two players won the top score (50 points) in the finals, each player needs to perform an extra slam dunk until champion is occurs. If one competitor wants to win the top score, he needs to show aesthetic in the take-off phase and flight phase of one slam dunk, because aesthetic was considered by referees as well as the difficulty in the assessment. As a result, the aesthetic in slam dunk plays a vital role in winning the championship during NBA slam dunk contest, which not exists in the NBA regular season [15]. ”

Comment 7. INTRODUCTION: The introduction is consistent and easy to follow. Hypotheses are clearly formulated.

OUR REPLY: Thank you for your encouragement for our study. We will make great efforts to improve the quality and preciseness based on the editor and reviewers’ suggestions.

Comment 8. INTRODUCTION: The Authors should clarify the actual heritage of this study. I am concerned about the originality of the present study.

OUR REPLY: Thank you for your helpful comments. We make corresponding explanation and amendment, based on your suggestion. As follows:

1. For the actual heritage of this study, we clarified that the slam dunk-related indicator (height without shoes, standing reach, weight, three quarter court sprint, standing vertical jump, and max vertical jump) was found in this study. Through the anthropometric and physical fitness data, the staff of NBA can select suitable rookies to slam dunk contest. However, coach from NBA or other basketball league, who want to improve the player’s slam dunk performance, will apply specific training programs to develop the slam dunk-related indicator. For example, coach can pay more attention on the three quarter courtsprint, standing vertical jump, max vertical jump, which will increase the efficiency of training program aimed at slam dunk performance. Furthermore, excellent slam dunk performance will increase the audience rating of basketball league and attract more people to like and join the basketball. At the same time, one successful on-court slam dunk will enhance the sports confidence of basketball players by promotion of functionality appreciation, because slam dunk is aesthetic and ornamental. One recent research provided relevant evidence that functionality appreciation was significantly correlated with sports confidence (p<0.036) (Ricketts C, Malete L, Myers ND, Bateman AG, Bateman CJ. Sport bodies: An examination of positive body image, sport-confidence, and subjective sport performance in Jamaican athletes. Psychology of Sport & Exercise. 2023; 67: 102434). To sum up, this study contributes to enhancing slam dunk performance, audience rating of basketball league and sports confidence of basketball players. 

2. The originality of the study is that the NBA players with good performance in slam dunk contest has shorter height without shoes and standing reach, less weight than eliminated player, which is the contrary result to previous studies. Previous viewpoints always conclude that taller player with higher standing reach has better basketball-related performance, because the distance between them and the rim is shorter. Interestingly, our result reflects that the finalist possessed lighter weight, compared to their peers. This is also opposite to the previous conclusion, reporting that basketball player with larger weight and less body fat percentage have better on-court performance. To sum up, compared to previous studies, current study came up with different results, reflecting the originality of this study, because the slam dunk performance is entirely different from other basketball-related performance.

3. We also made the corresponding revision to clarify the actual heritage of this study as follows:

(1) Introduction section:

“Thus, research assessing the basketball players, needs to propose the novel insights about players in the slam dunk contest to define their own characteristic in anthropometry and physical fitness. These insights allow the basketball coaches from NBA or other basketball league to develop the slam dunk performance of players, based on the slam dunk-related physical fitness, contributing to player’s sports confidence [35] and basketball league’s audience rating. Identification of potential differences on anthropometric and physical fitness indicator between different performances in the slam dunk contest (finalist vs eliminated player) can inform the slam dunk-related indicator for selecting the suitable participants into the contest, which can maximize its aesthetic [18] and audience rating.”

 Conclusions section:

“To conclude, specific anthropometric and physical fitness indicator is clearly different between finals group and elimination group. Height without shoes, standing reach, weight in anthropometry and three quarter court sprint, standing vertical jump, max vertical jump in physical fitness are key indicator to slam dunk performance. In accordance with the result in the study, NBA staff can select suitable rookies for slam dunk contest. Similiarly, coach from NBA or other basketball league, who want to improve the player’s slam dunk performance, should use specific training programs to develop the slam dunk-related indicator.”

 “application of the study” section in Discussion section:

“The application of the study is that, through slam dunk-related anthropometric and physical fitness indicator in this study, the NBA staff can select suitable rookies for slam dunk contest. Similarily, coach from NBA or other basketball league, who want to improve the player’s slam dunk performance, should use specific training programs to develop the slam dunk-related indicator. Furthermore, by enhancing on-court slam dunk performance, this study contributes to the audience rating for basketball league, and sports confidence of basketball players [35].”

Comment 9. METHOD: How was sample size determined? (Sampling technique!)

OUR REPLY: Thank you very much for pointing out. We made explanation on sampling technique as follows:

In first round sampling, totally 49 NBA players were selected from the NBA combine draft database to this study. Inclusion criterion is participant of NBA all star slam dunk contest during 2003-2023 season and combine draft during 2000-2015 season. Exclusion criterion is undrafted player during 2000-2015 season and non-participant of NBA slam dunk contest during 2003-2023 season. 

A total of 32 NBA players passed the screening as the subject and 17 players was excluded in second round sampling. Inclusion criterion is NBA player with complete draft data. Exclusion criterion is player without complete draft data. Because 5 indicators was excluded in the study, complete draft data indicates having complete data in the rest of indicators. Finally, based on the result of slam dunk contest, the subjects were divided into finals group (FG) and eliminated group (EG). The criterion of grouping is whether to enter the finals. The process of sampling is displayed in following figure.

Comment 10. METHOD: Ethical Consideration: Make sure you have the proper informed consent statement in the paper, i.e., subjects were informed of the experimental risks and signed an informed consent document prior to the investigation.

OUR REPLY: Thank you very much for pointing out. We have made corresponding explanation and revision:

The study is non-human subject research, analyzing the already available data from the NBA official draft database (www.nba.com/stats/draft/combine). Therefore, we can not directly communicate with NBA players, whose draft data was analyzed in this study. Previous studies, which employed available data from NBA draft database, also not provide the informed consent statement. We listed the researches using same method as follows:

1. Teramoto M, Cross CL, Rieger RH, Maak TG, Willick SE. Predictive validity of National Basketball Association draft combine on future performance. Journal of Strength & Conditioning Research.2018;32(2):396-408.

2. Cui Y, Liu F, Bao D, Liu H, Zhang S, Gómez M. Key anthropometric and physical determinants for different playing positions during National Basketball Association draft combine test. Frontiers in Psychology.2019;10:2359.

3. Zhang S, Lorenzo A, Gómez MA, Mateus N, Gonçalves B, Sampaio J. Clustering performances in the NBA according to players' anthropometric attributes and playing experience. Journal of Sports Sciences. 2018;36(22): 2511-2520.

4. Mehran N, Williams PN, Keller RA, Khalil LS, Lombardo SJ, Kharrazi FD.Athletic performance at the national basketball association combine after anterior cruciate ligament reconstruction. Orthopaedic Journal of Sports Medicine. 2016; 4(5): 2325967116648083.

5. Igor R, Radivoj M, Marko C, Predrag B, Milivoj D. NBA Pre-Draft Combine is the weak predictor of rookie basketball player’s performance. Journal of Human Sport and Exercise.2020:16(3):1-10.

On other hand, it is unquestionable that before the combine draft, participants was informed all the detailed information of draft, although we did not find that informed consent document of NBA players was made public in various sources. The reason refer to that the participant tried their best to practice for good performance in the combine draft, and thus they must clearly understand its information and agree to join the combine draft. 

For example, one report indicated that several participants of draft did pre-draft workout for a long time (www.nba.com/pacers/news/athletic-foward-prosper-prospering-in-pre-draft-setting). 

Thus, based on your helpful suggestions, we revised the “ethical consideration” section in Materials and Methods section as follows:

 “This study was non-human subject research, suggesting that all the data of this study were available data from the NBA official draft database [8-10]. All subjects in the study participated in the NBA combine draft on the basis of the informed consent.”

Comment 11. STATISTICAL ANALYSIS：Please, present methods of data analysis and criterion of results interpretation.

OUR REPLY: Thank you very much for pointing out. We have presented methods of data analysis and criterion of results interpretation. Our explanation is presented as follows:

 Descriptive statistics: Descriptive statistics was used to all the data in the study. It was presented as mean and standard deviation (Mean ± SD).

2. Normal distribution test: Shapiro–Wilks analysis was used for testing the normal distribution of data.

Criterion of results interpretation: the significance level of normal distribution was set at P＞0.05, indicating that data meet the normal distribution.

3. Homogeneity of variables test: Levene test was applied to examine the homogeneity of variables.

Criterion of results interpretation: the significance level of homogeneity of variables was set at P＞0.05, suggesting that data meet the homogeneity of variables.

4. T-test: independent sample t-test was employed to test the statistical significance of intergroup difference, while the data met the normal distribution and the homogeneity of variables.

Criterion of results interpretation: The significance level of independent sample t-test was set at P＜0.05.It reflected the statistical significance of intergroup difference.

5. Effect size: Cohen’s d level of independent sample t-test was employed as effect size to illustrate the extent of intergroup difference.

Criterion of results interpretation: The effect size was interpreted as: Cohen’s d<0.3, trivial; 0.3≤Cohen’s d<0.7, small; 0.7≤Cohen’s d<1.3, moderate;1.3≤Cohen’s d< 2.1, large; 2.1≤Cohen’s d<4.0, very large; Cohen’s d≥4.0, extreme large. It suggests the extent of intergroup difference between Finals Group and Eliminated Group.

To clarify the methods of data analysis and criterion of results interpretation, we also revised the “statistical analysis” section in Materials and Methods section as follows:

“At first, descriptive statistics was adopted for all the data in the study. It was presented as mean and standard deviation (Mean ± SD). Next, Shapiro–Wilks analysis was adopted for testing the normal distribution of data. Levene test was used to examine the homogeneity of variables. The independent sample t-test was employed to test the statistical significance of intergroup difference, while the data met the normal distribution and the homogeneity of variables.”

“Cohen’s d level was employed as effect size to illustrate the extent of intergroup difference and interpreted as: cohen’s d<0.3, trivial; 0.3≤cohen’s d<0.7, small; 0.7≤cohen’s d<1.3, moderate; 1.3≤cohen’s d<2.1, large; 2.1≤Cohen’s d<4.0, very large; cohen’s d≥4.0, extreme large [13-14]. The significance level of normal distribution and homogeneity of variables was set at P＞0.05. In addition, the statistical significance of independent sample t-test was set at P＜0.05. The statistical analysis was performed using SPSS Statistics 27.0 (IBM Corp, Armonk, New York, USA).”

Comment 12. STATISTICAL ANALYSIS：Please define specificity and sensitivity in this research context.

OUR REPLY: Thank you very much for pointing out. We made corresponding explanation:

1. Specificity: All indicators in this study show relatively high specificity to basketball-related performance. The reason is that these indicators often used for assessment and research of basketball-related performance in related studies. They are basketball-specific indicators, which can highly indicate basketball-specific performance, for NBA combine draft. The related study is presented as follow:

1. Teramoto M, Cross CL, Rieger RH, Maak TG, Willick SE. Predictive validity of National Basketball Association draft combine on future performance. Journal of Strength & Conditioning Research.2018;32(2):396-408.

2. Cui Y, Liu F, Bao D, Liu H, Zhang S, Gómez M. Key anthropometric and physical determinants for different playing positions during National Basketball Association draft combine test. Frontiers in Psychology.2019;10:2359.

3. Zhang S, Lorenzo A, Gómez MA, Mateus N, Gonçalves B, Sampaio J. Clustering performances in the NBA according to players' anthropometric attributes and playing experience. Journal of Sports Sciences. 2018;36(22): 2511-2520.

4. Mehran N, Williams PN, Keller RA, Khalil LS, Lombardo SJ, Kharrazi FD.Athletic performance at the national basketball association combine after anterior cruciate ligament reconstruction. Orthopaedic Journal of Sports Medicine. 2016; 4(5): 2325967116648083.

5. Igor R, Radivoj M, Marko C, Predrag B, Milivoj D. NBA Pre-Draft Combine is the weak predictor of rookie basketball player’s performance. Journal of Human Sport and Exercise.2020:16(3):1-10.

2. Sensitivity: According to the result in this study, height without shoes, standing reach, weight in anthropometry and three quarter court sprint, standing vertical jump, max vertical jump in physical fitness test has high sensitivity for explaining the slam dunk performance, due to the significant intergroup difference (P<0.05). On the contrary, the sensitivity of body fat percentage, wingspan and lane agility time was relatively low, because intergroup difference was not of statistical significance in these indicators (P>0.05).

Comment 13. RESULTS: Results description is a little chaotic and insufficient.

OUR REPLY: Your suggestion is greatly appreciated. Unfortunately, we were unable to add more result through new indicators, because this study employed the available data in the NBA draft data (www.nba.com/stats/draft/combine). 

Based on your suggestion, we made corresponding revision in Result section. At first, we have rewritten the Result sections based on the revised data, because we found the mistakes on Table 4 and Table 5. Next, we made the process of Result sections in good order, which was present as fixed format (result of descriptive statistics→result of independent sample t-test→result of effect size). Finally, we presented each result displayed in the tables. As follows:

 “The descriptive statistics for anthropometric indicators was displayed in Table 4. Independent sample t-test suggested that FG had significantly lower values than EG onheight W/O shoes, SR, and weight (P＜0.05), while the intergroup difference on BFP and wingspan were not of statistical significance (P>0.05) (see Table 4). ES test about cohen’s d demonstrated that extent of intergroup difference was moderate on height W/O shoes, SR, and weight (0.7≤cohen’s d<1.3), while it was small on BFP and wingspan(0.3≤cohen’s d<0.7) (see Table 5).”

“The descriptive statistics for physical fitness indicators was shown in Table 4. The result of independent sample t-test showed that FG has significant better performance than EG on TQCS, SVJ, and MVJ (P＜0.05), while there was no significant intergroup difference on LAT(P>0.05) (see Table 4).ES test about cohen’s d suggested that extent of intergroup difference was moderate on TQCS, SVJ and MVJ (0.7≤cohen’s d<1.3), whereas it was small on LAT(0.3≤cohen’s d<0.7) (see Table 5).”

Comment 14. RESULTS: Please, add some introductions to the description of the results and indicate what and why you did. Each result presented in the tables should be commented on in the text.

OUR REPLY: Your suggestion is greatly appreciated. At first, we have rewritten the Introduction to meet your suggestion. Then, based on your suggestion, we indicated what and why we did in the Result section and also presented each result displayed in the tables as follows:

 What we did: independent sample t-test.

Why we did: test the statistical significance of intergroup difference.

 What we did: Effect size analysis (Cohen’s d level of independent sample t-test).

Why we did: analyze the extent of intergroup difference.

 Our revision in the Results section is as follows:

“The descriptive statistics for anthropometric indicators was displayed in Table 4. Independent sample t-test suggested that FG had significantly lower values than EG onheight W/O shoes, SR, and weight (P＜0.05), while the intergroup difference on BFP and wingspan were not of statistical significance (P>0.05) (see Table 4). ES test about cohen’s d demonstrated that extent of intergroup difference was moderate on height W/O shoes, SR, and weight (0.7≤cohen’s d<1.3), while it was small on BFP and wingspan(0.3≤cohen’s d<0.7) (see Table 5).”

“The descriptive statistics for physical fitness indicators was shown in Table 4. The result of independent sample t-test showed that FG has significant better performance than EG on TQCS, SVJ, and MVJ (P＜0.05), while there was no significant intergroup difference on LAT(P>0.05) (see Table 4).ES test about cohen’s d suggested that extent of intergroup difference was moderate on TQCS, SVJ and MVJ (0.7≤cohen’s d<1.3), whereas it was small on LAT(0.3≤cohen’s d<0.7) (see Table 5).”

Comment 15. DISCUSSION：The discussion needs to reflect what you found, how it relates to the literature and then what it means physiologically or from a practical aspect and each paragraph should be logical in sequence as at present it is a bit hard to follow.

OUR REPLY: Thank for your helpful suggestions. Based on your valuable suggestions, we have rewritten the Discussion section. First, we discussed the anthropometric indicators in one paragraph, while we mentioned the physical fitness indicators in another paragraph, because all indicators in this study belonged to anthropometry or physical fitness. Then, we have organized the inter-paragraph logic, intra-paragraph logic and logic between indicators in Discussion section. The logical structure of Discussion section is presented as follows:

 First Paragraph

 The result of this study.

 The hypothesis of this study.

 Comparison between the result and the hypothesis in the study.

 Second Paragraph for anthropometric indicators.

 The result of indicator 1 (what we found).

 Comparison between the result in this study and previous studies (how it relates to the literature).

 Explanation of possible reason and the underlying principle (what it means physiologically or from a practical aspect).

 Example related to the result in this study.

 Discussion for indicator 2 (same as indicator 1).

 Discussion for indicator 3 (same as indicator 1).

 To sum up, indicator 1, 2, 3 was key anthropometric indicator for sports performance in the slam dunk contest. 

 Third Paragraph for physical fitness indicators (same as Second Paragraph).

 Fourth Paragraph is the application of this study.

 Last Paragraph is the limitation of this study.

Comment 16. DISCUSSION：Make sure the paper’s importance and need is clear to the reader.

OUR REPLY: Your suggestion is greatly appreciated. Based on your suggestion, we made the corresponding explanation and revision as follows:

 Importance of our study is following:

 We was clarified that the slam dunk-related indicator (height without shoes, standing reach, weight, three quarter court sprint, standing vertical jump, max vertical jump), which can reflect the slam dunk performance, was found in present study. 

 Through the slam dunk-related indicator that we found in current study, the staff of NBA or other basketball league can select suitable rookies to slam dunk contest, which is contribute to audience rating. 

 Coach from NBA or other basketball league, who want to improve the player’s slam dunk performance, can employ specific training programs to promote the slam dunk-related indicator that we found in current study. In this way, good slam dunk performance will improve the sports confidence of basketball players, because aesthetics in slam dunk was belonged to functionality appreciation. One recent study reported that functionality appreciation was significantly associated with sports confidence in athletes (p<0.036) (Ricketts C, Malete L, Myers ND, Bateman AG, Bateman CJ. Sport bodies: An examination of positive body image, sport-confidence, and subjective sport performance in Jamaican athletes. Psychology of Sport & Exercise. 2023; 67: 102434). 

 Need of our study is following:

 Staffs from NBA or other basketball league want to select the potential players for slam dunk contest and to improve the audience rating of the league.

 Basketball coaches from NBA or other league demand the development of the slam dunk performance of basketball players.

 Basketball players need improve their sports confidence.

 Thus, we made corresponding revisions as follows:

 “application of the study” section in Discussion section:

“The application of the study is that, through slam dunk-related anthropometric and physical fitness indicator in this study, the NBA staff can select suitable rookies for slam dunk contest. Similarily, coach from NBA or other basketball league, who want to improve the player’s slam dunk performance, should use specific training programs to develop the slam dunk-related indicator. Furthermore, by enhancing on-court slam dunk performance, this study contributes to the audience rating for basketball league, and sports confidence of basketball players [35].”

(2) Introduction section:

“Thus, research assessing the basketball players, needs to propose the novel insights about players in the slam dunk contest to define their own characteristic in anthropometry and physical fitness. These insights allow the basketball coaches from NBA or other basketball league to develop the slam dunk performance of players, based on the slam dunk-related physical fitness, contributing to player’s sports confidence [35] and basketball league’s audience rating. Identification of potential differences on anthropometric and physical fitness indicator between different performances in the slam dunk contest (finalist vs eliminated player) can inform the slam dunk-related indicator for selecting the suitable participants into the contest, which can maximize its aesthetic [18] and audience rating.”

Comment 17. CONCLUSION: Thus, my biggest concern is that the practical and cases of this particular study are not particularly high in the coach after reading this is not really thinking about changing any of their approaches to training or testing. Our main goal was not to provide changing of approaches to training or testing.

OUR REPLY: Thank you very much for pointing out. According to your helpful suggestions, we made the corresponding explanation and amendment to clarify the importance of this study as follows:

1. From the aspect of training, basketball coaches, who read our study, will comprehend the slam dunk-related physical fitness indicator (three quarter sprint, standing vertical jump and max vertical jump). If they want to improve the slam dunk performance of players, they can enhance the slam dunk-related indicator by specific training program, such as plyometric training (PT) with or without weight, velocity-based training (VBT), and post-activation potentiation (PAP).If coaches arrange the training program only based on their experience or studies about other basketball-related performance, they will discover many possible influence factors for slam dunk, including flexibility, postural stability, speed of approach and take-off, dribbling technique, flight height and distance. Although our research may not directly influence the approach of training, it improves the efficiency of slam dunk training, because our study pointed out what is the most important for slam dunk performance.

2. From the aspect of testing, this study revealed that slam dunk-related indicator, to coaches or staffs from NBA or other basketball league. As a result, they will select the most suitable players to improve the aesthetic and quality of the slam dunk contest. For example, they can invite the basketball players into dunk contest, who has short height without shoes and standing reach, light weight with good performance on three quarter sprint, standing vertical jump and max vertical jump. In addition, our study also discovered the specific test plan for selecting the slam dunk players, including the measurement of height without shoes, standing reach, weight in anthropometry and three-quarter sprint, standing vertical jump and max vertical jump in physical fitness test.

3. We made corresponding revisions as follows:

(1) Conclusions section:

“To conclude, specific anthropometric and physical fitness indicator is clearly different between finals group and elimination group. Height without shoes, standing reach, weight, in anthropometry and three quarter court sprint, standing vertical jump, max vertical jump in physical fitness are key indicator to slam dunk performance. In accordance with the result in the study, NBA staff can select suitable rookies for slam dunk contest. Similiarly, coach from NBA or other basketball league, who want to improve the player’s slam dunk performance, should use specific training programs to develop the slam dunk-related indicator.”

(2) “application of this study” section in Discussion section:

“The application of the study is that, through slam dunk-related anthropometric and physical fitness indicator in this study, the NBA staff can select suitable rookies for slam dunk contest. Similarily, coach from NBA or other basketball league, who want to improve the player’s slam dunk performance, should use specific training programs to develop the slam dunk-related indicator. Furthermore, by enhancing on-court slam dunk performance, this study contributes to the audience rating for basketball league, and sports confidence of basketball players [35].”

(3) Introduction section:

“Thus, research assessing the basketball players, needs to propose the novel insights about players in the slam dunk contest to define their own characteristic in anthropometry and physical fitness. These insights allow the basketball coaches from NBA or other basketball league to develop the slam dunk performance of players, based on the slam dunk-related physical fitness, contributing to player’s sports confidence [35] and basketball league’s audience rating. Identification of potential differences on anthropometric and physical fitness indicator between different performances in the slam dunk contest (finalist vs eliminated player) can inform the slam dunk-related indicator for selecting the suitable participants into the contest, which can maximize its aesthetic [18] and audience rating.”

Comment 18. CONCLUSION: Why might one want to cite this paper? What is the true impact of the literature?

OUR REPLY: Your suggestion is greatly appreciated. Our corresponding explanation and revision is as follows:

1. The strength and conditioning researchers will cite the study. One possible reason is that according to the evidence in our study, they can find more slam dunk-related indicators for different types of subjects (national-level basketball athlete, state-level basketball players, collegiate-level players, and slam dunk players from slam dunk clubs).Another reason is that they will be able to design the specific training program for slam dunk performance, because our study indicates the key indicators from physical fitness, which plays a vital role in the slam dunk training. 

2. The researchers in the field of basketball training will cite the study, because based on the result of our study, they can explore more key indicators, which is associated with various tasks of on-court slam dunk, such as slam dunk with or without defense, slam dunk with dribbling, and alley-oop (dunk without dribbling). Our work discovered the slam dunk-related physical fitness, and thus basketball researchers, who read our study, also can develop specific training program to improve the performance of on-court slam dunk.

3. Researchers from the field of sports biomechanics may read our study. The reason refers to that there was potential new mechanism of biomechanics (kinematics: different joint angle and angular velocity, kinetics: different vertical ground reaction force, human material mechanics: different vertical stiffness, joint stiffness, stress and strain) in slam dunk, a complex technique combination in different situation. In addition, it attracts them to analyze the movement of slam dunk.

4. The true impact of this study contained the academic and practical aspects. At academic level, our study will provide relevant evidence to researchers in the strength and conditioning, basketball training and sports biomechanics. The practical impact refers to that this study will contribute to slam dunk performance, audience rating of basketball league and sports confidence of basketball players. 

5. Based on your valuable suggestions, we made the corresponding revisions as follows:

(1) Conclusions section:

“To conclude, specific anthropometric and physical fitness indicator is clearly different between finals group and elimination group. Height without shoes, standing reach, weight, in anthropometry and three quarter court sprint, standing vertical jump, max vertical jump in physical fitness are key indicator to slam dunk performance. In accordance with the result in the study, NBA staff can select suitable rookies for slam dunk contest. Similiarly, coach from NBA or other basketball league, who want to improve the player’s slam dunk performance, should use specific training programs to develop the slam dunk-related indicator.”

(2) “application of this study” section in Discussion section:

“The application of the study is that, through slam dunk-related anthropometric and physical fitness indicator in this study, the NBA staff can select suitable rookies for slam dunk contest. Similarily, coach from NBA or other basketball league, who want to improve the player’s slam dunk performance, should use specific training programs to develop the slam dunk-related indicator. Furthermore, by enhancing on-court slam dunk performance, this study contributes to the audience rating for basketball league, and sports confidence of basketball players [35].”

(3) Introduction section:

“Thus, research assessing the basketball players, needs to propose the novel insights about players in the slam dunk contest to define their own characteristic in anthropometry and physical fitness. These insights allow the basketball coaches from NBA or other basketball league to develop the slam dunk performance of players, based on the slam dunk-related physical fitness, contributing to player’s sports confidence [35] and basketball league’s audience rating. Identification of potential differences on anthropometric and physical fitness indicator between different performances in the slam dunk contest (finalist vs eliminated player) can inform the slam dunk-related indicator for selecting the suitable participants into the contest, which can maximize its aesthetic [18] and audience rating.”

Reviewer#3:

Comment 1. The limitations of the study, both internally and externally, are considerable. However, the authors present them in the discussion. Considering the use of already available data (and not collected specifically for the study), the limitations are understandable.

OUR REPLY: Thank you very much for pointing out. In the future study, we will collect the first-hand data from participants of NBA all-star slam dunk contest to break through the limitations. For instance, we will apply the high-speed camera and motion capture system to collect the kinematics of approach phase and flight phase during the contest and also perform new anthropometry and physical fitness test with comprehensive indicators. 

Comment 2. Abstract: The penultimate sentence of the abstract is a bit confusing and long. Please try to make it more direct and easily understandable.

OUR REPLY: Your suggestion is greatly appreciated. We rewrote this sentence and made it more direct and easily understandable as follows:

“To conclude, specific anthropometric and physical fitness indicator shows clear difference between finals group and elimination group.”

Comment 3. Introduction: Page 9, line 4 – the term "contained" seems to me not to be the most appropriate, please review.

OUR REPLY: Thank you for your helpful suggestion. We changed the term “contained” to “used”, aiming to illustrate that National Basketball Associational also uses the anthropometric measurement and physical fitness test in the draft. Our revision is presented as follows:

“Anthropometric measurement and physical fitness test was used in the National Basketball Association (NBA) combine draft for selecting the potential NBA players [4].”

Comment 4. Introduction: Page 9, lines 18-19 – The 1st sentence of this 2nd paragraph is confusing. Please specify clearly the audience referred to and the sporting season to which it refers. Or does this audience refer to all seasons between 2013-2018?

OUR REPLY: We appreciate it very much for this good suggestion, and we have made the corresponding amendments and explanations:

The audience that we referred to is television audience from Los Angeles, USA. The sporting season is all-star week before play-off season from 2003 to 2018. Therefore, for clear expression, we revised this sentence to clarify the audience and sporting season as follows:

“NBA all star game has great attraction to basketball audience, during all star weekend from 2003 to 2018, with up to totally 659.7 television audiences in 1000 family in Los Angeles, USA [7].”

Comment 5. Introduction: The authors talk about the "draft" and "slam dunk contest" as if the reader is already fully familiar with this selection and competition process. Given that this manuscript may be of interest to a wider variety of readers, including colleagues not as familiar with these NBA events, it is advisable to introduce them in the introduction.

OUR REPLY: Thank you for your helpful comments. We agreed entirely and made the corresponding revisions as follows:

1. First, we added the introduction of NBA combine draft, including the test content, requirement of participants, and the meaning of draft sequence as follows:

“NBA combine draft includes anthropometric measurement, physical fitness test, spot-up shooting test and non-stationary shooting test and scrimmages. All rookies, who participated in the combine draft, must be at least 19 years old, with experience in other basketball league [3-4]. Draft sequence indicates the rank of rookies, which was selected early or lately by NBA team. Therefore, rookies, who rank in the top sequence, always attract more attention in the future session [2].”

2. Next, we introduced the requirement of participants, competition process and rule in NBA all star slam dunk contest, and tried to illustrate them in the clearest way as follows:

“Though all the rookies with less than three year of NBA experience were allowed to apply for the contest, the number of competitors was kept between three and six [34]. All the competitors drew lots to decide their order of play before the preliminary. After one player completed a slam dunk in limit time, each referee should give one individual score from six to ten, while the sum of all referee’s score is the total score of one slam dunk. Each referee can give six as individual score, if one competitor failed to make one slam dunk in required time. Competitors, who ranked in top two or top three in preliminary, can reach the finals. When both of two players won the top score (50 points) in the finals, each player needs to perform an extra slam dunk until champion is occurs. If one competitor wants to win the top score, he needs to show aesthetic in the take-off phase and flight phase of one slam dunk, because aesthetic was considered by referees as well as the difficulty in the assessment. As a result, the aesthetic in slam dunk plays a vital role in winning the championship during NBA slam dunk contest, which not exists in the NBA regular season [15]. ”

Comment 6. Introduction: Page 10, line 7 – the authors refer to reference number 48, however the list of references only goes up to number 46. Please check.

OUR REPLY: We appreciate it very much for this good suggestion. We changed this reference number from “number 48” to “number 33”, because we deleted some references in the study, which changed the sequence in references section. As follows:

33. Markwick WJ, Bird SP, Tufano JJ, Seitz LB, Haff GG. The intraday reliability of the reactive strength index calculated from a drop jump in professional men's basketball. International Journal of Sports Physiology & Performance. 2015; 10(4): 482-488.

Comment 7. Introduction: Page 10, lines 2-8 – The authors present anthropometric characteristics and references that support slam dunk performance in basketball players, however some of these references refer to the sport of volleyball (ref 32 and 43), another to athletes in general (ref 34) and another even more general reference (45). Please reconsider if all the references in this sentence are well selected.

OUR REPLY: Thank you for your helpful suggestion. We made corresponding explanation and revision:

The main reason is that evidence about this topic (the relationship between physical fitness and slam dunk performance in basketball players) is seldom from various sources. Therefore, in this study, we only found some reports about other sports and subjects to discuss the association between some physical fitness indicators and slam dunk performance in basketball players. For the preciseness of reference, we have removed “reference number 32, 34, 43, 45” from Introduction section as follows:

“Finalist in the contest probably possesses optimal body fat percentage [3-4], height and standing reach [3-4, 16, 24, 36], weight [3-4, 16, 24, 36],wingspan [3-4] in the anthropometry, and extremely outstanding performance on linear sprint [3-4, 16, 37], standing vertical jump and countermovement jump [24, 27, 29], max vertical jump [3-4, 24, 27, 29], and reactive strength index (RSI) [33] in the physical fitness test, supporting their slam dunk-specific performance. ”

However, we have cited them in the Discussion section. The detailed reason is presented as follows:

1. Reference number 32 and 43 in discussion explained the association between standing vertical jump and max vertical jump in basketball players, because spike jump and attack jump in volleyball and double-legs take-off without dribbling in the slam dunk shared similar take-off technique (both of them are double legs without dribbling, and the slow stretch-shorten cycle).

2. Reference number 34 in discussion supported the association between standing vertical jump and 20-30 three quarter court meter sprint, because both of the standing vertical jump and countermovement jump belonged to slow stretch-shorten cycle with double-legs take-off, while three quarter court meter sprint (22.86 meter sprint) has similar distance with linear sprint test (20 meter, 25 meter, 30 meter) in reference number 34.

3. Reference number 45 in discussion provided one possible influence factor (length of feet and toes) of max vertical jump.

Comment 8. Introduction: Page 10, lines 14-16 – The hypothesis regarding the anthropometric component is misplaced. The authors state that they expect "that finalist has greater value in the anthropometry", please clarify which anthropometric variables you expect to find this "greater value" in. Anthropometry is the study of the physical measurements and dimensions of the human body, not "the measurements themselves". By saying "greater value in the anthropometry" the reader can interpret for example greater body mass, or greater amount of fat mass, which does not seem to me to be the hypothesis raised.

OUR REPLY: Thank you for your helpful comments. We have clarified that compared to eliminated player, finalist has taller height without shoes, higher standing reach, longer wingspan and lower body fat percentage in the anthropometry with performance in the physical fitness test. In line with your question and suggestion, we have rewritten the hypothesis to meet your requirement as follows: 

“The hypothesis of the present study refers to that finalist was possessed taller height without shoes and standing reach, longer wingspan, and lower body fat percentage with better performance in the physical fitness test, in relative to eliminated player.”

Comment 9. Materials and methods：Table 1 – The last variable presented is "Score Mean (point)", the legend of this variable is unclear, are we talking about average points during the competition? Average of the various competitions? Please clarify.

OUR REPLY: Thank you for your helpful comments. We have made the corresponding explanations: 

Score Mean (point) indicates the average point for one player within one competition. We have removed this indicator, which not belong to anthropometric indicator or physical fitness indicator, from Table 1, because it has less relationship to the key topic that we discussed in the study as follows:

Comment 10. Materials and methods：Page 11, Participants – The authors say the sample consisted of 49 participants. However, due to the criteria they present, they had to eliminate 17, so the final sample and, subject to statistical treatment was only 32 athletes. In other words, the sample of the present study is 32 athletes and not 49, so this information should be clarified both in the "sample" section and in the abstract.

OUR REPLY: Thank you very much for pointing out. Based on your valuable suggestion, we revised the “participants” section in Materials and Methods section and Abstract section:

(1) “participants” section in Materials and Methods section:

“The combine draft data of 32 NBA players from NBA official draft database was sampled as participants in the present study（N=32, age in draft year: 20.69±2.28 years old, height without shoes: 196.75±8.68 cm, weight: 96.85±10 kg, body fat percentage: 6.07±1.23%）[8-10]. Table 1 displays participants’ characteristics.”

“Inclusion criterion for subjects in this study: The subject has participated in at least once NBA all star slam dunk contest during 2003-2023 season and combine draft during 2000-2015 season. The subject has complete draft data. Exclusion criterion was undrafted player during 2000-2015 season, non-participant of NBA slam dunk contest during 2003-2023 season, and player with incomplete draft data.”

(2) Abstract section: 

“Draft data of 32 basketball players (N=32, age in draft year: 20.69±2.28 years old, height without shoes: 196.75±8.68 cm, weight: 96.85±10 kg, body fat percentage: 6.07±1.23%) participating in the 2000-2015 draft and 2003-2023 slam dunk contest was selected from national basketball association database.”

Comment 11. Materials and methods: Page 12, Antropometrics Indicators – Information regarding the assessment of anthropometric characteristics is insufficient. The authors report that measurements were performed "by the NBA conditioning coach by valid and reliable protocol". Can you ensure that these coaches were certified anthropometrists? If they were not, these measurements are no longer scientifically valid, as there was no control over measurement error of anthropometric variables. Please clarify this topic. What anthropometric protocols were used? A brief description that does not fully clarify the protocols used is presented in Table 2. For example, the description given for the variable "height" does not mention the positioning of the head. According to the ISAK (International Society for the Advancement of Kinathropometry) protocol, the head would be positioned in the Frankfurt plane. Please complement and support with references, which anthropometric protocols are used.

OUR REPLY: Thank you very much for pointing out. We have made corresponding explanation and revision:

1. We tried our best to search the anthropometric protocol of NBA combine draft and profile of NBA strength and conditioning coach through all sources, while we did not find the detailed description of what we need. In addition, we discovered that most of studies about anthropometry in NBA combine draft, share the unified standard of anthropometric protocol. One reason is that all the studies in this research field cited and analyzed the anthropometric data in same database (https://www.nba.com/stats/draft/combine-anthro). Another reason refers to that all studies about the anthropometry of NBA combine draft were reported the same protocol (www.topendsports.com/sport/basketball testing-nba-draft.htm). The related studies are presented as follow:

1. Teramoto M, Cross CL, Rieger RH, Maak TG, Willick SE. Predictive validity of National Basketball Association draft combine on future performance. Journal of Strength & Conditioning Research.2018;32(2):396-408.

2. Cui Y, Liu F, Bao D, Liu H, Zhang S, Gómez M. Key anthropometric and physical determinants for different playing positions during National Basketball Association draft combine test. Frontiers in Psychology.2019;10:2359.

3. Zhang S, Lorenzo A, Gómez MA, Mateus N, Gonçalves B, Sampaio J. Clustering performances in the NBA according to players' anthropometric attributes and playing experience. Journal of Sports Sciences. 2018;36(22): 2511-2520.

4. Mehran N, Williams PN, Keller RA, Khalil LS, Lombardo SJ, Kharrazi FD.Athletic performance at the national basketball association combine after anterior cruciate ligament reconstruction. Orthopaedic Journal of Sports Medicine. 2016; 4(5): 2325967116648083.

5. Igor R, Radivoj M, Marko C, Predrag B, Milivoj D. NBA Pre-Draft Combine is the weak predictor of rookie basketball player’s performance. Journal of Human Sport and Exercise.2020:16(3):1-10.

2. The anthropometric indicator is insufficient in the study, in relative to relevant research. The reason refers to that we used available data from NBA draft database in the study. It is hard to add more anthropometric indicators. However, most of subjects missed the data on width of hand, length of hand and height with shoes, because the number of subjects in the study is significant lower than previous studies, which means participants of NBA all star slam dunk contest is few. In addition, there was also a difficulty of studying the performance in the slam dunk contest. In this study, there is nothing we can do to introduce new anthropometric indicators, because the data in this study was already available data. We will add more indicators in the subsequent study if we have chance to perform new anthropometry for NBA players.

3. Based on your suggestion, we made a new Table for anthropometric protocol and revised the information as follows:

Comment 12. Materials and methods: Statistical Analysis – The authors applied one-way ANOVA, noting that both groups had normal distribution of the data. Another assumption for the application of this statistical test is the homogeneity of variables. Was this assumption tested?

OUR REPLY: We appreciate it very much for this good suggestion. We have made corresponding amendments and explanations: 

As presented in following table, the data in the study was tested in the Levene test and met the homogeneity of variables, which indicated that P value is larger than 0.05. Based on the suggestions, we added the Levene test in the “statistical analysis” section of Materials and Methods section as follows:

 “At first, descriptive statistics was adopted for all the data in the study. It was presented as mean and standard deviation (Mean ± SD). Next, Shapiro–Wilks analysis was adopted for testing the normal distribution of data. Levene test was used to examine the homogeneity of variables. The independent sample t-test was employed to test the statistical significance of intergroup difference, while the data met the normal distribution and the homogeneity of variables.”

Tests of Homogeneity of Variances

　 Levene Statistic df1 df2 P value

BODY FAT PERCENTAGE(%) Based on Mean 0.277 1 30 0.603

 Based on Median 0.201 1 30 0.657

 Based on Median and with adjusted df 0.201 1 22.172 0.658

 Based on trimmed mean 0.218 1 30 0.644

HEIGHT W/O SHOES (cm) Based on Mean 0.868 1 30 0.359

 Based on Median 0.463 1 30 0.502

 Based on Median and with adjusted df 0.463 1 24.212 0.503

 Based on trimmed mean 0.586 1 30 0.450

STANDING REACH (cm) Based on Mean 0.175 1 30 0.679

 Based on Median 0.164 1 30 0.688

 Based on Median and with adjusted df 0.164 1 26.147 0.689

 Based on trimmed mean 0.168 1 30 0.685

WEIGHT (kg) Based on Mean 0.521 1 30 0.476

 Based on Median 0.839 1 30 0.367

 Based on Median and with adjusted df 0.839 1 26.831 0.368

 Based on trimmed mean 0.558 1 30 0.461

WINGSPAN(cm) Based on Mean 0.003 1 30 0.956

 Based on Median 0.020 1 30 0.889

 Based on Median and with adjusted df 0.020 1 26.959 0.889

 Based on trimmed mean 0.001 1 30 0.970

LANE AGILITY TIME (s) Based on Mean 0.625 1 30 0.435

 Based on Median 0.597 1 30 0.446

 Based on Median and with adjusted df 0.597 1 29.687 0.446

 Based on trimmed mean 0.564 1 30 0.458

THREE QUARTER SPRINT 

(s) Based on Mean 0.001 1 30 0.982

 Based on Median 0.001 1 30 0.982

 Based on Median and with adjusted df 0.001 1 26.661 0.982

 Based on trimmed mean 0.004 1 30 0.949

STANDING VERTICAL LEAP 

(cm) Based on Mean 0.018 1 30 0.893

 Based on Median 0.018 1 30 0.894

 Based on Median and with adjusted df 0.018 1 29.914 0.894

 Based on trimmed mean 0.018 1 30 0.893

MAX VERTICAL LEAP 

(cm) Based on Mean 1.215 1 30 0.279

 Based on Median 0.987 1 30 0.329

 Based on Median and with adjusted df 0.987 1 28.124 0.329

 Based on trimmed mean 1.173 1 30 0.287

Comment 13. Results: Page 14, Results, lines 4-5 – Consider replacing the phrase "was significantly less than" by "had significantly lower values than".

OUR REPLY: Thank you for your helpful comments. According to your valuable suggestion, we have changed this phrase “was significantly less than" to "had significantly lower values than" as follows:

 “Independent sample t-test suggested that FG had significantly lower values than EG onheight W/O shoes, SR, and weight (P＜0.05), while the intergroup difference on BFP and wingspan were not of statistical significance (P>0.05) (see Table 4).”

Comment 14. Results: Table 2 – Please add the acronym of each test to the table, it can help the reader to “incorporate” the acronym.

OUR REPLY: Your suggestion is greatly appreciated, and we have added the acronym of each test in Table 2 and Table 3 as follows:

Comment 15. Discussion：Page 15, lines 3-4 – Please reconsider the presentation of the hypothesis regarding anthropometry data, in line with the previous suggestion.

OUR REPLY: Thank you for your helpful comments. We have revised the hypothesis in Discussion section, in line with the previous suggestion as follows:

“It hypothesized that finalist was possessed taller height without shoes and standing reach, longer wingspan, and lower body fat percentage with better performance in the physical fitness test, in relative to eliminated player.”

Comment 16. Discussion：Page 15, line 19 – Please replace “Figure II” by “Figure 2.”. Is the image not copyrighted? Do you have permission to use it?

OUR REPLY: Thank you very much for pointing out. We are unclear that whether Figure 2 is copyrighted or not. Therefore, we have removed the Figure 2, because we had no permission to use it.

Comment 17. Discussion：Page 15, lines 18-22 – Not all readers may know that aesthetic is accounting in slam dunk. It is really important that this information is clear in the introduction and can be reinforced when presented in the discussion.

OUR REPLY: Thank you for your helpful comments. First, we have added this information that aesthetic is accounting in slam dunk in the Introduction section as follows:

“If one competitor wants to win the top score, he needs to show aesthetic in the take-off phase and flight phase of one slam dunk, because aesthetic was considered by referees as well as the difficulty in the assessment. As a result, the aesthetic in slam dunk plays a vital role in winning the championship during NBA slam dunk contest, which not exists in the NBA regular season [15]. ”

However, we reinforced this information in the Discussion section as follows:

“One possible reason refers to the difference of on-court performance between slam dunk contest and NBA regular season. In slam dunk contest, aesthetic and difficulty of air-technique was key on-court performance, while was not vital to regular season [15]. Especially the aesthetic, it is a determinant of top score in the contest, but is unimportant to score in regular season.”

Comment 18. Discussion：Page 16, lines 1-2 – Please clarify how a shorter limb (arm or lower extremity) can produce a higher greater velocity. In this actual form, it seems that you are saying that a shorter arm produces a higher angular velocity. I supposed that isn´t it that you want to say, maybe you are talking about the tendon insertion?

OUR REPLY: We appreciated it very much for this good suggestion, and we have made the corresponding explanations as follows:

When the linear velocity is constant, athlete with shorter limbs will generate the greater angular velocity in motion. The reason refers to that angular velocity (ω) is equal to linear speed (v) divided by arm of force (r), which is typical formula of kinematics in the biomechanics (ω=v/r). If one athlete has short limbs, it suggests that the short arm of force, which contributes to greater angular velocity. For example, elite gymnastics athlete often possesses short height and standing reach, because their short arm of force will result in the large angular velocity during the twist and somersault of air-technique. 

In addition, larger angular velocity in the take-off phase may be beneficial for slam dunk-specific MVJ. One biomechanics study reported that the flight height of spike jump in volleyball, which share similar jump characteristics (e.g. slow stretch-shorten cycle, double legs include lead leg and trail leg, orientation step in take-off phase) with one type of slam dunk-specific MVJ (double-legs take-off), was positively correlated with the maximal angular velocity of dominant knee extension in the take-off phase (p<0.001, r=0.85) (Fuchs PX, Fusco A, Bell JW, von Duvillard SP, Cortis C, Wagner H. Movement characteristics of volleyball spike jump performance in females. Journal of Science & Medicine in Sport. 2019; 22(7): 833-837). It revealed an increase in jump height by 1 cm per 22° s−1 increase in maximal dominant knee extension angular velocity. As a result, short height and standing reach contributes to flight height of slam dunk-related MVJ, because of the greater angular velocity of dominant knee extension in the take-off phase.

Based on your suggestion, we have revised the Discussion section as follows:

“In this way, it was likely that FG has better performance than EG on aesthetic and difficulty of air-technique, due to the discrepancies on shorter height and SR between groups. It can be explained by the greater angular velocity. As shown in the formula (ω=v/r), when linear speed (v) is similar between groups, FG will generate greater angular velocity (ω) than EG, because of their shorter arm of force (r), which positively correlated with height and SR (P＜0.05) [17]. As a result, greater angular velocity may contribute to the number and velocity of twist, which likely to influence the difficulty and aesthetic of air-technique. For example, short height was extensively found in elite athlete from rhythmic gymnastics. It was highly correlated performance score (P＜0.001, r=0.8), which also scored based on the difficulty and aesthetic of air-technique [18-19]. In addition, greater joint angular velocity, which was resulted from shorter height and SR, may also be benificial for slam dunk-related MVJ. It has been reported that maximal angular velocity of dominant knee extension in take-off phase was positively correlated with spike jump in volleyball (P<0.001, r=0.85) [22], which has similar jump characteristics (slow stretch-shorten cycle, double legs include lead leg and trail leg, orientation step) with one type of slam dunk-specific MVJ (double-legs take-off without dribbling) [21-22, 27]. As a result, if FG finished one slam dunk by this type of MVJ, their flight height will be higher than EG that also used it, which may also lead to the intergroup difference on difficulty and aesthetic of air-technique.”

Comment 19. Discussion：Page 16, line 13 – Please replace “dead mass” by fat mass, we all hope that participant do not have dead mass.

OUR REPLY: Your suggestion is greatly appreciated, and we have replaced “dead mass” by “fat mass” as follows:

“It suggested that FG, who owned the superior slam dunk performance, may has greater lean body mass (LBM) and less fat mass, because its BFP is also less than EG in current study, aside from weight. Similar to the current study, Cui Y et al. reported that fat mass and LBM had influence on SVJ and MVJ in NBA players (P<0.01), which was likely to determine the jump performance on slam dunk”

Comment 20. Discussion：Page 18, line 1 – Please linguistically revise this 1st line.

OUR REPLY: Thank you for your helpful comments. This sentence was deleted, because we have rewritten the Discussion section.

---

## [Decision Letter · Decision Letter 1]

29 Jan 2024

PONE-D-23-19002R1Anthropometric and physical fitness indicators in the combine draft between the finalist and the eliminated player in the national basketball association all-star slam dunk contestPLOS ONE

Dear Dr. Tse-hau Tong,

Thank you for submitting your manuscript to PLOS ONE. After careful consideration, we feel that it has merit but does not fully meet PLOS ONE’s publication criteria as it currently stands. Therefore, we invite you to submit a revised version of the manuscript that addresses the points raised during the review process.

**ACADEMIC EDITOR: **Dear authors,

The authors answered almost all the questions made by reviewers. While one reviewer already recommended acception, the other still found some issues in materials and methods.

Authors are welcome to answer. 

In addition, it is suggested replacing average by mean (see tables). P-values should use the same number of decimal (three). Please standardize them. 

Best regards

Please submit your revised manuscript by Mar 14 2024 11:59PM. If you will need more time than this to complete your revisions, please reply to this message or contact the journal office at plosone@plos.org. Please include the following items when submitting your revised manuscript:A rebuttal letter that responds to each point raised by the academic editor and reviewer(s). You should upload this letter as a separate file labeled 'Response to Reviewers'.A marked-up copy of your manuscript that highlights changes made to the original version. You should upload this as a separate file labeled 'Revised Manuscript with Track Changes'.An unmarked version of your revised paper without tracked changes. You should upload this as a separate file labeled 'Manuscript'.If applicable, we recommend that you deposit your laboratory protocols in protocols.io to enhance the reproducibility of your results. Protocols.io assigns your protocol its own identifier (DOI) so that it can be cited independently in the future. For instructions see: https://journals.plos.org/plosone/s/submission-guidelines#loc-laboratory-protocols. Additionally, PLOS ONE offers an option for publishing peer-reviewed Lab Protocol articles, which describe protocols hosted on protocols.io. Read more information on sharing protocols at https://plos.org/protocols?utm_medium=editorial-email&utm_source=authorletters&utm_campaign=protocols.

We look forward to receiving your revised manuscript.

Kind regards,

Rafael Franco Soares Oliveira

Academic Editor

PLOS ONE

Journal Requirements:

Additional Editor Comments:

Dear authors,

The authors answered almost all the questions made by reviewers. While one reviewer already recommended acception, the other still found some issues in materials and methods.

Authors are welcome to answer.

In addition, it is suggested replacing average by mean (see tables). P-values should use the same number of decimal (three). Please standardize them.

Best regards

Reviewers' comments:

Reviewer's Responses to Questions

**Comments to the Author**

1. If the authors have adequately addressed your comments raised in a previous round of review and you feel that this manuscript is now acceptable for publication, you may indicate that here to bypass the “Comments to the Author” section, enter your conflict of interest statement in the “Confidential to Editor” section, and submit your "Accept" recommendation.

Reviewer #1: All comments have been addressed

Reviewer #3: All comments have been addressed

2. Is the manuscript technically sound, and do the data support the conclusions?

Reviewer #1: Yes

Reviewer #3: Yes

3. Has the statistical analysis been performed appropriately and rigorously? 

Reviewer #1: Yes

Reviewer #3: Yes

4. Have the authors made all data underlying the findings in their manuscript fully available?

Reviewer #1: Yes

Reviewer #3: Yes

5. Is the manuscript presented in an intelligible fashion and written in standard English?

Reviewer #1: Yes

Reviewer #3: Yes

6. Review Comments to the Author

Reviewer #1: First of all, the reviewer would like to thank the authors for their work and efforts in trying to improve sports science knowledge. Overall, the study is well designed and well-written, with a great original article evaluating the usefulness of the

topic

Accepted

Reviewer #3: General Comments

The authors answered all the questions and made a considerable effort to follow all the suggestions for improvement. The current version of the manuscript is clearer and more complete than the previous one, and I congratulate the authors on their work.

Nevertheless, there are one aspect that still need improvement or clarification, which I'll explain below.

Materials and methods

Anthropometric Indicators - Unfortunately the information on the protocol used is still missing. The authors have identified a link to the protocols, but when I click on the link I can't identify it.

All the questions I raised in Round 1 about the validity of anthropometric data remain. Extracting data from the same database doesn't fully guarantee that:

- the same protocol has always been followed,

- the data was collected by certified anthropometrists.

These issues represent clear threats to the internal validity of the study and should, at the very least, be clearly presented and discussed in the limitations section/paragraph.

7. PLOS authors have the option to publish the peer review history of their article (what does this mean?). If published, this will include your full peer review and any attached files.

Reviewer #1: No

Reviewer #3: No

---

## [Author Response · Author response to Decision Letter 1]

4 Feb 2024

Dear Editor and Reviewers:

Thank you for your letter and for the reviewers’ comments concerning our manuscript entitled “Anthropometric and physical fitness indicators in the combine draft between the finalist and the eliminated player in the national basketball association all-star slam dunk contest” (Manuscript Number: PONE-D-23-19002). Those comments are all valuable and very helpful for revising and improving our paper, as well as the important guiding significance to our research. We have studied comments carefully and have made corrections which we hope to meet with approval. We highlighted all revision in red color. The corrections in the paper and the responds to the referee’s comments are as following:

Additional Editor Comments：

Comment 1. The authors answered almost all the questions made by reviewers. While one reviewer already recommended acception, the other still found some issues in materials and methods.

OUR REPLY: Thank for helpful comments. We have tried best to revise this manuscript. However, we have changed affiliation of the first author with the permission of the editor in Plos One. The reason is that the first author already have enrolled in Harbin Institute of Technology (HIT) as Phd student, and left previous affiliation. 

Comment 2. In addition, it is suggested replacing average by mean (see tables).

OUR REPLY: Thank for your helpful comments. According to your suggestion, we have replaced “average” by “mean” in Table 1 and Table 4. Our revision is following:

Comment 3. P-values should use the same number of decimal (three). Please standardize them.

OUR REPLY: Thank for your helpful comments. Based on your suggestion, we have amended them to same number of decimal. As follows:

Reviewer#1: 

Comment 1.First of all, the reviewer would like to thank the authors for their work and efforts in trying to improve sports science knowledge. Overall, the study is well designed and well-written, with a great original article evaluating the usefulness of the

topic

OUR REPLY: Thank for your encouragement. We have tried our best to revise the manuscript, aiming to meet the requirement of Plos One.

Reviewer#3:

Comment 1. The authors answered all the questions and made a considerable effort to follow all the suggestions for improvement. The current version of the manuscript is clearer and more complete than the previous one, and I congratulate the authors on their work. Nevertheless, there are one aspect that still need improvement or clarification, which I'll explain below.

OUR REPLY: Thank for your encouragement. We have spared no effort to amend this manuscript to meet the requirement of Plos One.

Comment 2. Materials and methods:

Anthropometric Indicators - Unfortunately the information on the protocol used is still missing. The authors have identified a link to the protocols, but when I click on the link I can't identify it. All the questions I raised in Round 1 about the validity of anthropometric data remain. Extracting data from the same database doesn't fully guarantee that:

- the same protocol has always been followed,

- the data was collected by certified anthropometrists.

These issues represent clear threats to the internal validity of the study and should, at the very least, be clearly presented and discussed in the limitations section/paragraph.

OUR REPLY: Thank for your helpful comments. We have tried our best to search any useful information that can answer this question from various sources. Unfortunately, we failed to seek out the detailed protocol and test staff’s personal information for NBA combine draft. At present, we can not directly contact the NBA manager, who was in charge of NBA combine draft, for more information. Therefore, We deemed that it is not possible to confirm with certainty that a uniform protocol was followed by certified anthropometrists. In our future study, the standard and detailed protocol was designed and followed by certified anthropometrists for internal validity. 

According to your suggestions, we have added one section in the limitations paragraph and ranked it first, clearly discussing this aspect. As follows:

However, this study still has the following limitations. First, although the anthropometric data in NBA combine draft proposes valuable insights, there were certain variations in test protocol and anthropometrist’s qualification, which restricted the internal validity of this study. Second, some NBA draft indicators was excluded, due to the large number of missing value [4]. These indicators were not discussed in this study. Besides, the data in this study was anthropometry and physical fitness test in the draft year, which was not the physical condition during slam dunk contest, so sports injury or other disease may occur in the season after draft [4]. However, RSI, exhibiting good reliability on the assessment of jump performance for male basketball players, was not used in the physical fitness test, and it will be used in the future study [33]. Finally, the number of subjects was comparatively low in the study, because only 3-6 rookies were allowed to join the NBA all star slam dunk contest every year [34]. In the future research, members from slam dunk clubs in the world, such as Team Flight Brother (TFB) in USA, Dunk Elite (DE) in Europe, AirChina (AC) and Dunk King (DK) in China, were recruited as the subjects.

---

## [Editor Report · Decision Letter 2]

7 Feb 2024

Anthropometric and physical fitness indicators in the combine draft between the finalist and the eliminated player in the national basketball association all-star slam dunk contest

PONE-D-23-19002R2

Dear Dr. Tse-hau Tong,

We’re pleased to inform you that your manuscript has been judged scientifically suitable for publication and will be formally accepted for publication once it meets all outstanding technical requirements.

Kind regards,

Rafael Franco Soares Oliveira

Academic Editor

PLOS ONE

Additional Editor Comments (optional):

Dear authors,

You have completed in a proper way all requests made by me or reviewers.

Therefore, my recommendation is to accept!

Congratulations!

Best regards
---

## [Editor Report · Acceptance letter]

21 Feb 2024

PONE-D-23-19002R2 

PLOS ONE

Dear Dr. Tong, 

I'm pleased to inform you that your manuscript has been deemed suitable for publication in PLOS ONE. Congratulations! Your manuscript is now being handed over to our production team.

Kind regards, 

on behalf of

Prof Rafael Franco Soares Oliveira 

Academic Editor

PLOS ONE